# Enhancing CLIP with CLIP: Exploring Pseudolabeling for Limited-Label Prompt Tuning

**Cristina Menghini**
Brown University
cristina_menghini@brown.edu

**Andrew Delworth**
Brown University
adelwort@cs.brown.edu

**Stephen H. Bach**
Brown University
sbach@cs.brown.edu

## Abstract

Fine-tuning vision-language models (VLMs) like CLIP to downstream tasks is often necessary to optimize their performance. However, a major obstacle is the limited availability of labeled data. We study the use of pseudolabels, i.e., heuristic labels for unlabeled data, to enhance CLIP via prompt tuning. Conventional pseudolabeling trains a model on labeled data and then generates labels for unlabeled data. VLMs' zero-shot capabilities enable a "second generation" of pseudolabeling approaches that do not require task-specific training on labeled data. By using zero-shot pseudolabels as a source of supervision, we observe that learning paradigms such as semi-supervised, transductive zero-shot, and unsupervised learning can all be seen as optimizing the same loss function. This unified view enables the development of versatile training strategies that are applicable across learning paradigms. We investigate them on image classification tasks where CLIP exhibits limitations, by varying prompt modalities, e.g., textual or visual prompts, and learning paradigms. We find that (1) unexplored prompt tuning strategies that iteratively refine pseudolabels consistently improve CLIP accuracy, by 19.5 points in semi-supervised learning, by 28.4 points in transductive zero-shot learning, and by 15.2 points in unsupervised learning, and (2) unlike conventional semi-supervised pseudolabeling, which exacerbates model biases toward classes with higher-quality pseudolabels, prompt tuning leads to a more equitable distribution of per-class accuracy. The code to reproduce the experiments is at BatsResearch/menghini-neurips23-code.

## 1 Introduction

Large pre-trained vision-language models (VLMs) [31, 43, 17] achieve remarkable accuracy without task-specific training but still require adaptation for optimal performance. Prompt-tuning [13, 18] is an approach to efficiently enhance VLMs performance on downstream tasks by learning inputs to the model. While learning prompts with a few labeled data can yield significant improvements [48, 2], a broader range of learning settings such as semi-supervised, transductive zero-shot, and unsupervised learning are still underexplored. All of these settings share access to unlabeled data, and the versatile zero-shot classification abilities of VLMs make pseudolabeling a natural approach to leveraging it. This paper investigates how the use of out-of-the-box pseudolabels assigned by CLIP can contribute to improving CLIP's own performance. To this end, we conduct an extensive exploration of learning scenarios by varying prompt modalities, learning paradigms, and training strategies. We present empirical evidence showcasing the effectiveness of iterative prompt-training strategies that leverage CLIP-based pseudolabels, regardless of learning paradigms and prompt modalities, resulting in significant improvements in CLIP's image classification performance across different settings.

Pseudolabels are heuristic labels assigned by a model to unlabeled data, which are leveraged to further train the model [20]. Successful training with pseudolabels relies on two factors: the quality

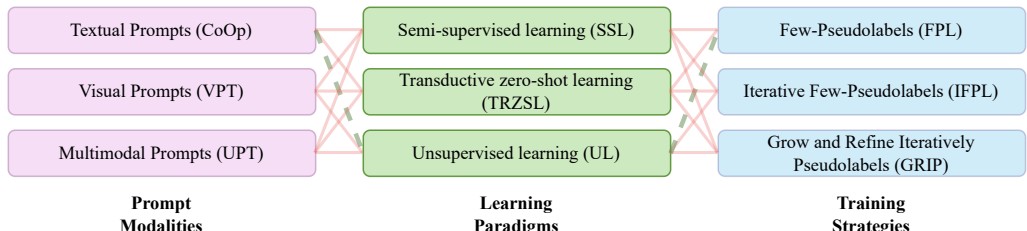

| Prompt
Modalities | Learning
Paradigms | Training
Strategies |
|---|---|---|

Figure 1: Our design space to explore the effect of leveraging pseudolabels in a unified way across prompt modalities, learning paradigms, and training strategies. The green (dashed) path has already been explored [15], while the red (solid) lines are the unexplored combinations for prompt tuning.

of the labels and how they are used during training. To address the first, conventional methods assign labels to instances with high-confidence predictions [36]. For pseudolabeling using CLIP, Huang et al. propose to select the most confident samples for each class [15], mitigating CLIP's bias [38] and miscalibration [22] (see Section 3). To assign pseudolabels, we rely on this approach and address the second point by exploring how to make the best use of them. We design a broad space of analysis considering three dimensions: prompt modalities, which are the model inputs we learn; learning paradigms, which define the data we have available; and training strategies, which describe the process used to optimize performance (Figure 1).

Research on prompt tuning has demonstrated that training strategies used for learning prompts in one modality can be transferred to learning prompts in a different modality. For instance, Visual Prompt Tuning [18] was originally designed to effectively fine-tune large vision models but can be adapted to efficiently fine-tune CLIP using the same training strategy as standard textual prompt tuning [48, 34, 44]. On the contrary, different learning paradigms with limited labeled data typically require distinct approaches specifically tailored to extract information from the available data [27, 12]. However, we observe that this changes by using VLM's generated pseudolabels. Unlike conventional pseudolabeling approaches that bootstrap off labeled data and are used as semi-supervised learning techniques [36, 3, 40], VLMs can generate pseudolabels in any learning setting. This offers a significant advantage, expanding the scope of pseudolabeling beyond semi-supervised learning, and making it a promising approach for other settings, such as transductive zero-shot and unsupervised learning. By using CLIP-based pseudolabels as a source of supervision, we can view these settings as optimizing the same loss function, which is simply a weighted sum of the errors on labeled data, if available, and pseudolabeled data. Given that we can express different settings as the same problem, we can propose training strategies, i.e., the way of using pseudolabels, that suit them all.

By standardizing the training strategies across various prompt modalities and learning settings, we can conduct experiments on different applications of pseudolabels for various combinations of prompt modalities, learning paradigms, and training strategies, as illustrated in Figure 1. To the best of our knowledge, only one potential path has been explored thus far; specifically, fine-tuning textual prompts in an unsupervised learning context using a few pseudolabels [15]. Rather than relying on a fixed set of pseudolabels, we propose iterative training techniques that allow for the ongoing refinement and expansion of the pool of pseudolabeled data used during training. With each iteration, we progressively enhance CLIP's pseudolabeling ability, allowing us to extend the set of pseudolabeled data while maintaining the high quality of the initial pseudolabels.

We conduct experiments on six tasks where CLIP has been observed to underperform [31], such as satellite-image classification, flower-species identification, and texture-image recognition, among others. Our findings reveal that iterative approaches effectively fine-tune prompts irrespective of their modality and learning paradigms. Recent studies have identified the "Matthew effect" as a potential issue for semi-supervised models that use pseudolabels [49, 38]. This phenomenon causes models to perform well on classes with accurate pseudolabels but poorly on those with inaccurate ones, thereby reinforcing the model's original bias towards certain classes. Our analysis reveals that using pseudolabels generated by CLIP for prompt-tuning with iterative strategies not only improves CLIP's overall performance but also corrects its natural bias towards certain classes.

We summarize the main takeaways of our work:

- General purpose zero-shot learners used as general purpose pseudolabelers open the opportunity to develop training strategies that leverage pseudolabeled data beyond semi-supervised learning. We point out that different learning paradigms, such as semi-supervised, transductive zero-shot, and unsupervised learning, can be all considered as special cases of a single objective function, by using pseudolabels as a source of supervision.

- We demonstrate that simple iterative training strategies for refining pseudolabels are highly effective approaches for limited-label prompt tuning. In fact, regardless of the prompt modality and learning setting, these strategies improve CLIP, by on average 19.5 points in semi-supervised learning, 28.4 in transductive zero-shot learning, and 15.2 in unsupervised learning.

- We show that prompts learned with iterative strategies help mitigate the "rich get richer, poor get poorer" effect observed in semi-supervised approaches leveraging pseudolabels. By redistributing the quality of pseudolabels across different classes, we observe a "Robin Hood effect" where the extremely rich classes' accuracy stays the same or decreases, while poorer classes get richer, leading to a more equitable distribution of per-class accuracy.

## 2 Background and related work

**Vision-language models**  Vision-language models such as CLIP [31], ALIGN [17], and Florence [43] are models that align images and text. We focus on CLIP, which is composed of two components: a text encoder, $\psi$, and an image encoder, $\phi$, which are jointly trained using a contrastive loss to learn a multi-modal embedding space which aligns the representations of similar text and image inputs. This pre-training enables CLIP to perform zero-shot image classification. Given an image $x$ and a set of classes $\mathcal{Y} = \{y_1, ..., y_C\}$, CLIP classifies $x$ by measuring the similarity between the image representation $z = \phi(x)$ and each class representation $w_i = \psi(\pi_i)$, based on their cosine distance in the shared embedding space. Here, $\pi_i$ is a natural language prompt such as ``a photo of a [CLASS$_i$]'', where CLASS$_i$ is the specific class name, such as "orange dahlia," "forest" or "Boeing 737". The image $x$ gets assigned to the class with the highest similarity score. In this work, we study how to learn better prompts that enhance CLIP by leveraging pseudolabels.

**Prompt tuning**  Prompt tuning is a technique that enhances the practical application of large pretrained models like CLIP [31] and GPT [32, 7]. It involves providing task-specific information to the model during inference through textual or visual inputs, leading to improved performance on downstream tasks [1, 33, 6, 7]. While discrete prompts are manually crafted natural language descriptions of classes that guide the model, they may not yield optimal results [47]. Soft prompting [21, 24], on the other hand, optimizes prompts as continuous vectors. These can be optimized by backpropagating through the frozen pre-trained model, resulting in better performance. Soft prompts can be learned for various modalities, e.g., text or image, [48, 13, 17, 2, 44, 19] and applications [34, 27, 12, 28] by training on a small number of labeled examples per class. If only unlabeled data is accessible, it is possible to learn textual soft prompts by leveraging CLIP-based pseudolabels [15]. Expanding on this concept, we further investigate the use of pseudolabels across a broader range of prompt modalities and learning approaches, and we introduce unexplored training strategies to leverage pseudolabels more effectively.

**Learning from pseudolabels**  Pseudolabeling is the practice of assigning labels to unlabeled data based on the prediction of a model [20]. Then, pseudolabels are used to improve the performance of the model itself. There are different ways to obtain and use pseudolabels and each impacts the final predictions of the model [41, 45, 16, 35]. Some approaches use confidence thresholds [36, 3, 40] and others average predictions from multiple augmentations [4]. Pseudolabeling is a semi-supervised learning technique, and it is rarely used in transductive zero-shot learning [42, 5, 25]. Applying such techniques requires a few labeled examples related to the target task to learn a baseline model capable of pseudolabeling. However, this limitation has been overcome by VLMs, which are capable of pseudolabeling examples without task-specific training. The conventional pseudolabeling scheme based on confidence threshold is not effective if we assign pseudolabels based on CLIP. In fact CLIP is miscalibrated [22] and has imbalanced predictions [38] which may induce noise in the pseudolabels. An alternative approach selects the top-K most confident examples per class to improve performance [15]. In our analysis, we rely on this scheme (Section 3).

# 3 Design space

Our analysis encompasses the design space consisting of various combinations of prompt modalities, learning paradigms, and training strategies (Figure 1). Within this space, two key components remain constant: the pseudolabeling scheme and a unified loss function. This section begins by introducing these components and subsequently delves into a comprehensive discussion of each dimension within the design space to be explored.

**Pseudolabeling scheme**    The use of CLIP to generate pseudolabels has been investigated in [15]. Given unlabeled data $X_u$ with target classes $\{y_1, ..., y_C\}$, the goal is to assign labels to data points in which the model is most confident. Typically, pseudo labeling schemes use a confidence threshold $(P(y|x) > \tau)$ to select instances to pseudolabel. However, this approach does not work well for CLIP due to its miscalibration [22] and imbalanced predictions [38]. Instead, one can use a top-K pseudo labeling approach, where the top-K most confident examples per class are used as pseudolabeled data [15]. The pseudolabel assignment consists of (1) computing the similarity scores of each datapoint with classes' textual prompts, and (2) select for each class the $K$ datapoints with the highest similarity score to the class. In this way, we always get $K$ pseudolabels per class, effectively addressesing the natural bias in CLIP's pseudolabels [38]. In Appendix A.1, we provide more details about pseudolabel assignment corner cases.

This top-K pseudolabeling scheme is applicable to unlabeled data, regardless of the availability of labeled data. As a result, we can extend the use of pseudolabels to any learning setting that involves unlabeled data. We observe that by treating pseudolabeled examples as true labeled data, we can view all learning settings as optimizing the same objective function.

**Unified objective function**    Consider a $C$-class image classification task, where $X_L$ and $Y_L$ represent the image representations and labels of the labeled data, and $X_U$ and $\tilde{Y}_U$ denote the image representations and pseudolabels for the unlabeled data. We define a loss function that combines two cross-entropy losses, one accounting for the error on the labeled data points and the other accounting for the error on pseudolabeled data:

$$\mathcal{L} = \mathcal{L}_{CE}(X_L, Y_L) + \lambda \, \mathcal{L}_{CE}(X_U, \tilde{Y}_U) \tag{1}$$

where $\gamma$ and $\lambda$ define the training balance between the errors on labeled and pseudolabeled data.

## 3.1 Prompt modalities

Learning prompts is the process of training a set of vectors $\mathrm{P} = [\mathrm{p}]_1 \ldots [\mathrm{p}]_K$ that are prepended to the textual or visual inputs of the encoders within the CLIP architecture. By prepending these vectors to specific inputs, we can learn *textual* prompts, *visual* prompts, or *multimodal* prompts when applying a set of vectors to both inputs simultaneously. We provide a technical and detailed explaination in Appendix A.1.

In our exploration, we consider all three types of prompts. The efficacy of prompts can vary depending on the task. Text prompt tuning may be most beneficial when image features are well-separated by class but may not be aligned with the corresponding textual prompt. Visual prompts rearrange the image features within the projection space, and it has the potential to improve CLIP when the pre-trained image features are not well separated by class. Finally, multimodal prompts allows for beneficial interaction between the two separate modalities, which might lead to both separable visual features, and text classifiers that are well-aligned with the corresponding visual features.

## 3.2 Learning paradigms

By adjusting the values of parameters $\gamma$ and $\lambda$ and using the appropriate sets of labeled and pseudolabeled data, the unified objective loss can be customized for each learning paradigm. We note that the redundancy in the use of parameters is for notation clarity in descriptions of the different strategies below. One can simply use $\lambda$ as balancing factor between labeled and pseudolabeled data.

**Semi-supervised learning**   In the semi-supervised learning (SSL) scenario we have access to a limited number of labeled data for all the target classes $D_L = \{(x, y)\}$ where $x$ is an input feature and $y \in \mathcal{Y} = [C]$ is the corresponding label. In addition, we have access to unlabeled data $X_U = \{x\}$, where $x$ is an image in the target domain $\mathcal{Y}$. From $X_U$, we get $\mathcal{D}_{PL} = \{(x, \tilde{y})\}$, where $\tilde{y} \in [C]$ is $x$'s pseudolabel. When using the unified loss in this setting, we set $\gamma$ to $|\mathcal{D}_{PL}|/|\mathcal{D}_L|$. As $|\mathcal{D}_L|$ is much smaller than $|\mathcal{D}_{PL}|$, $\gamma$ acts as an upweighting factor for the few-labeled instances, thus counterbalancing the learning effect of pseudolabels ($\lambda$=1).

**Transductive zero-shot learning**   In transductive zero-shot learning (TRZSL), we are provided with labeled data $D_L = \{(x, y)\}$ for some target classes $S$ (referred to as *seen* classes), where $x$ represents input features, and $y \in [S]$ is the corresponding label. Additionally, we have access to unlabeled data $X_U = \{x\}$ for a disjoint set of classes $U$ (referred to as *unseen* classes). Using $X_U$, we obtain $\mathcal{D}_{PL} = (x, \tilde{y})$, where $\tilde{y} \in [U]$ denotes the pseudolabels for $x$. The value of $\lambda$ in the unified loss is set to $|\mathcal{D}_L|/|\mathcal{D}_{PL}|$, which makes the weight of the pseudolabel loss equivalent to that of the labeled data ($\gamma = 1$). This is necessary because an imbalance in the number of labeled and pseudolabeled samples can result in a skewed training distribution, leading to better performance on seen classes while the performance on unseen classes may either remain stagnant or degrade. Studying this setting is interesting beyond transductive zero-shot learning. In fact, it has the potential to generalize to scenarios where the target task involves unseen classes, while the seen classes consist of auxiliary labeled data from the same domain but different task [30].

**Unsupervised learning**   In the unsupervised learning (UL) setting, we have access only to unlabeled data $X_U = \{x\}$, from which we obtain $\mathcal{D}_{PL} = (x, \tilde{y})$, where $\tilde{y} \in [C]$ denotes the pseudolabel for $x$. In this case, $\gamma$ is set to 0, as there is no labeled data, and $\lambda = 1$. The use of this setting was initially explored in [15], who leveraged a few pseudolabels per class to learn textual prompts. In this paper, we build on their work by investigating a variety of training strategies and prompt modalities.

**Supervised learning**   In supervised learning (SL), we are only provided with labeled data $D_L = (x, y)$, where $x$ represents an input feature, and $y \in [C]$ is the corresponding label. If we set $\lambda$ to 0, the unified loss function is equivalent to the objective functions of default prompt-tuning approaches that optimize the prompts using a few labeled instances per target class. This setting is not strictly part of our design space. However, we will refer to it to define baselines in Section 4.

### 3.3   Training strategies

The unified objective function enables the development of training strategies broadly applicable across various learning paradigms. We explore three distinct learning strategies to effectively use pseudolabels in this context. The first strategy uses pseudolabels in a static manner. The other two strategies, which are unexplored for prompt tuning, involve the dynamic use of pseudolabeled data.

**Few-pseudolabels (FPL)**   We select $K$ pseudolabels per target class, resulting in a pseudolabeled dataset of size $K \cdot C$. We learn the prompts by minimizing the objective function via backpropagation through CLIP's encoders. This strategy aligns with Unsupervised Prompt Learning (UPL) in [15]. We refer to it as few-pseudolabels (FPL) to encompass its applicability for learning prompts of diverse modalities across learning paradigms.

**Iterative Refinement of FPL (IFPL)**   Similar to FPL, we obtain the top-K pseudolabels for each target class. These pseudolabels are then used to train a new task-specific prompt. After completing the training, we use the learned prompt to compute the top-K pseudolabels per class again. Subsequently, we reinitialize the prompt and repeat this entire process for a total of $I$ iterations. With this iterative approach, if training with the initial pseudolabel set leads to an improvement in the model's performance, the model itself can become a more effective pseudolabeler, refining the pseudolabels in each subsequent iteration.

**Grow and Refine Iteratively Pseudolabels (GRIP)**   Although IFPL can improve the quality of the $K \times C$ pseudolabels used for training, it still limits learning to a few examples per target class. To overcome this constraint, we explore a method similar to IFPL, but with a key difference. In each iteration, we progressively increase the value of $K$. Specifically, during the $i$-th iteration, we use

$K = (i \times \frac{|X_U|}{I})/C$ of the unlabeled data to perform the steps in the iterative process. GRIP maintains class balance by selecting the top-K samples at each iteration, with $K$ increasing progressively. Similar to IFPL, both prompts and pseudolabels are reinitialized with every iteration, in order to avoid accumulating errors from earlier iterations. In other words, learning progresses from pseudolabels to new prompts to new pseudolabels, and so on. The rationale behind this strategy is that as the model's accuracy in generating pseudolabels improves, we can increase the total number of pseudolabels without introducing excessive noise.

## 4 Experiments

We explore the design space outlined in Section 3 to understand the effectiveness of leveraging pseudolabels for limited-label prompt tuning. We show that (1) iterative strategies significantly improve CLIP's performance across prompt modalities and learning settings, (2) using CLIP-based pseudolabels with iterative strategies induces a more equitable distribution of per-class accuracy.

**Datasets**    We conduct the analysis on six tasks, covering specialized and fine-grained domains, where CLIP shows deficiencies [31]. We call this set of tasks FRAMED, and it includes **F**lowers102 [29], **R**ESICS45 [9], FGVC-**A**ircraft [26], **M**NIST [11], **E**uroSAT [14], and **D**TD [10]. For each dataset we use the training and test splits provided in [23]. For the transductive zero-shot learning setting we randomly generate three splits of seen and unseen classes with a 62-38 ratio. Further details are in Appendix A.2.

**Baselines**    To evaluate the effectiveness of the training strategies described in Section 3.3, we compare the performance of CLIP when queried with the learned soft prompts to CLIP zero-shot with default prompts such as "a photo of a [CLASS]." In addition, we compare with default supervised prompt-tuning baselines, for which we only use the available labeled data: CoOp [48] for textual prompts, VPT [18] for visual prompts, and UPT [44] for multimodal prompts. We defer to Appendix A.1 the technical details of these methods.

**Evaluation metrics**    We assess the performance of each method by measuring the accuracy of the test set, averaging the results over five runs. In the case of TRZSL, we report the harmonic of the accuracies of seen and unseen classes to account for their potentially imbalanced performance [39].

**Training settings**    For all experiments, datasets, and learning strategies, we use ViT-B/32 as the vision backbone. For both visual and textual prompt learning, we set the prefix size to 16 [48, 18]. Multimodal prompts have length 8 [44]. We use SGD as the optimizer and train for 150 epochs. We use 5 warmup epochs at a learning rate of 0.0001, and then set the learning rate to $l$, which is decayed by the cosine annealing rule. For textual and visual prompt learning, $l = 0.1$, while for multimodal prompt learning, $l = 0.01$. In SSL, we use 2 labeled samples per class to assess the impact of pseudolabels in the scenario of very few labeled data and abundant unlabeled data. The number of iterations $I$ is 10. FPL and IFPL have the number of pseudolabels per class fixed to 16 since it is indicated as the optimal $K$ in the previous research on pseudolabeling with CLIP [15]. In general, $K$ is a hyperparameter that may require optimization in practical cases. We decide to be consistent with the literature and apply this fixed value of $K$ in order to reduce the addition of more confounding factors in our analysis.

### 4.1 Exploring the design space

**GRIP consistently enhances CLIP across prompt modalities and learning settings**    Table 1 reports the performance of GRIP, the best performing among the training strategies in Section 3.3, compared to CLIP and prompt-tuning baselines. Overall, GRIP consistently improves the performance of CLIP and the baselines across prompt modalities and learning settings. By tuning textual prompts, the average improvement over CLIP is 20.7 points in SSL, 14.9 in UL, and 32.4 in TRZSL, while the improvement on CoOp is 9.6 points in SSL, and 26.6 in TRZSL. Similar results for the visual prompts show that GRIP improves CLIP by 18.2 points in SSL, 15.7 in UL, and 30.8 in TRZSL, and VPT by 12.9 points in SSL, and 20.8 in TRZSL. We note that CoOp and VPT applied to the SSL setting correspond to learning only on the labeled data, and we do not run them in the UL setting as

**Textual prompts**

| | Flowers102 | | | RESICS45 | | | FGVCAircraft | | |
|---|---|---|---|---|---|---|---|---|---|
| Method | SSL | UL | TRZSL | SSL | UL | TRZSL | SSL | UL | TRZSL |
| CLIP | $63.67_{0.00}$ | | $63.40_{0.00}$ | $54.48_{0.00}$ | | $54.46_{0.00}$ | $\mathbf{17.58}_{0.00}$ | | $17.86_{0.00}$ |
| CoOp | $76.76_{1.11}$ | - | $63.22_{0.02}$ | $58.53_{10.81}$ | - | $63.37_{0.02}$ | $14.91_{3.22}$ | - | $21.70_{0.03}$ |
| GRIP | $\mathbf{83.6}_{0.68}$ | $\mathbf{69.84}_{1.06}$ | $\mathbf{86.26}_{0.00}$ | $\mathbf{74.11}_{0.68}$ | $\mathbf{70.55}_{0.88}$ | $\mathbf{81.07}_{0.00}$ | $16.98_{0.82}$ | $15.22_{0.71}$ | $\mathbf{26.08}_{0.00}$ |
| Δ CLIP | ↑ 19.93 | ↑ 6.17 | ↑ 22.86 | ↑ 19.63 | ↑ 16.07 | ↑ 26.61 | ↓ 0.6 | ↓ 2.36 | ↑ 8.22 |
| Δ CoOp | ↑ 6.84 | - | ↑ 23.04 | ↑ 15.58 | - | ↑ 17.70 | ↑ 2.07 | - | ↑ 4.38 |
| | MNIST | | | EuroSAT | | | DTD | | |
| CLIP | $25.10_{0.00}$ | | $20.77_{0.00}$ | $32.88_{0.00}$ | | $30.54_{0.00}$ | $43.24_{0.00}$ | | $43.45_{0.00}$ |
| CoOp | $56.42_{2.66}$ | - | $21.15_{0.09}$ | $59.51_{4.55}$ | - | $49.68_{0.08}$ | $37.10_{5.45}$ | - | $46.3_{0.03}$ |
| GRIP | $\mathbf{71.78}_{3.59}$ | $\mathbf{67.88}_{2.76}$ | $\mathbf{74.06}_{0.00}$ | $58.66_{2.64}$ | $\mathbf{57.21}_{1.77}$ | $\mathbf{92.33}_{0.00}$ | $\mathbf{56.07}_{0.85}$ | $\mathbf{46.09}_{1.06}$ | $\mathbf{65.30}_{0.01}$ |
| Δ CLIP | ↑ 46.68 | ↑ 42.78 | ↑ 53.29 | ↑ 25.78 | ↑ 24.33 | ↑ 61.79 | ↑ 12.83 | ↑ 2.85 | ↑ 21.85 |
| Δ CoOp | ↑ 15.36 | - | ↑ 52.91 | ↓ 0.85 | - | ↑ 42.65 | ↑ 18.97 | - | ↑ 19.00 |

**Visual prompts**

| | Flowers102 | | | RESICS45 | | | FGVCAircraft | | |
|---|---|---|---|---|---|---|---|---|---|
| Method | SSL | UL | TRZSL | SSL | UL | TRZSL | SSL | UL | TRZSL |
| CLIP | $63.67_{0.00}$ | | $63.40_{0.00}$ | $54.48_{0.00}$ | | $54.46_{0.00}$ | $17.58_{0.00}$ | | $17.86_{0.00}$ |
| VPT | $63.73_{1.52}$ | - | $64.71_{0.00}$ | $60.80_{1.65}$ | - | $67.06_{0.00}$ | $17.76_{0.68}$ | - | $\mathbf{26.69}_{0.00}$ |
| GRIP | $\mathbf{67.95}_{1.2}$ | $63.09_{0.55}$ | $\mathbf{77.18}_{0.00}$ | $\mathbf{71.22}_{0.77}$ | $\mathbf{68.43}_{0.61}$ | $\mathbf{82.19}_{0.00}$ | $\mathbf{19.43}_{0.5}$ | $17.51_{0.61}$ | $26.42_{0.00}$ |
| Δ CLIP | ↑ 4.28 | ↓ 0.58 | ↑ 13.78 | ↑ 16.74 | ↑ 13.95 | ↑ 27.73 | ↑ 1.85 | ↓ 0.07 | ↑ 8.56 |
| Δ VPT | ↑ 4.22 | - | ↑ 12.47 | ↑ 10.42 | - | ↑ 15.13 | ↑ 1.67 | - | ↓ 0.27 |
| | MNIST | | | EuroSAT | | | DTD | | |
| CLIP | $25.10_{0.00}$ | | $20.77_{0.00}$ | $32.88_{0.00}$ | | $30.54_{0.00}$ | $43.24_{0.00}$ | | $43.45_{0.00}$ |
| VPT | $42.53_{14.13}$ | - | $25.51_{0.05}$ | $47.13_{1.34}$ | - | $62.24_{0.02}$ | $36.41_{2.17}$ | - | $44.16_{0.01}$ |
| GRIP | $\mathbf{69.66}_{5.51}$ | $\mathbf{68.04}_{1.11}$ | $\mathbf{69.54}_{0.01}$ | $\mathbf{63.48}_{3.09}$ | $\mathbf{63.68}_{3.42}$ | $\mathbf{96.97}_{0.00}$ | $\mathbf{54.57}_{4.86}$ | $\mathbf{50.51}_{0.99}$ | $\mathbf{62.78}_{0.00}$ |
| Δ CLIP | ↑ 44.56 | ↑ 42.94 | ↑ 48.77 | ↑ 30.60 | ↑ 30.80 | ↑ 66.43 | ↑ 11.33 | ↑ 7.27 | ↑ 19.33 |
| Δ VPT | ↑ 27.14 | - | ↑ 44.03 | ↑ 16.35 | - | ↑ 34.73 | ↑ 18.16 | - | ↑ 18.62 |

Table 1: For each learning paradigm, we compare the accuracy of GRIP with CLIP zero-shot (ViT-B/32), CoOp, and VPT. Results are for SSL, UL, and TRZSL on FRAMED. We average the accuracy on 5 seeds and report the standard deviation. Δ METHOD is the difference between the accuracy of GRIP and METHOD. We note that for UL we can not apply CoOp and VPT since no labeled data is available.

there is no labeled data. Results are similar for multimodal prompts. We defer them to Appendix A.3, due to space constraints.

**No prompt modality is clearly superior** Using pseudolabels dynamically is beneficial for each modality. However, determining the clear superiority of one prompt modality over the other is challenging, as it depends on the specific tasks. For example, visual prompts work better for EuroSAT, while textual prompts excel in Flowers102. Despite intuitive explanations (Section 3.1), the scientific consensus remains elusive [44]. Hence, we prefer to emphasize that the dynamic use of pseudolabels consistently improves performance for each prompt modality, without declaring one modality as definitively better than the other.

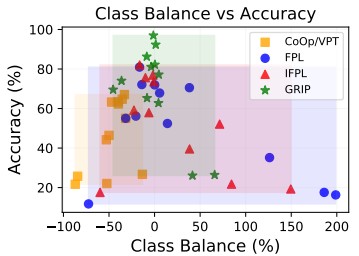

Figure 2: Balance of seen and unseen accuracies vs. model's overall accuracy. Points close to 0 indicate a good balance. Negatives represent better accuracy for the seen classes.

**Unsupervised learning is equivalent or more robust than learning with very few shots** The accuracy of GRIP when applied to the fully unsupervised setting is either higher or equivalent to the accuracy of VPT, which is trained using two labeled instances per class (Table 1). This shows that pseudolabeled data can substitute very few labeled examples for prompt tuning. However, the significant improvement of GRIP over CoOp and VPT in the semi-supervised setting (see Table 1) suggests that leveraging unlabeled data through pseudolabeling is advantageous in scenarios where labeled data is scarce but there is an abundance of unlabeled data.

| Textual prompts | Flowers102 | | | RESICS45 | | | DTD | | |
|---|---|---|---|---|---|---|---|---|---|
| Method | SSL | UL | TRZSL | SSL | UL | TRZSL | SSL | UL | TRZSL |
| FPL | $75.96_{0.74}$ | $65.67_{0.23}$ | $80.97_{0.00}$ | $68.13_{0.55}$ | $63.07_{0.38}$ | $72.11_{0.00}$ | $37.10_{5.45}$ | $44.96_{0.55}$ | $46.3_{0.03}$ |
| IFPL | $78.68_{0.75}$ | $\mathbf{69.56}_{1.05}$ | $82.08_{0.00}$ | $70.52_{1.12}$ | $64.11_{0.98}$ | $75.51_{0.00}$ | $\mathbf{55.24}_{0.97}$ | $\mathbf{47.77}_{1.15}$ | $59.14_{0.02}$ |
| GRIP | $\mathbf{83.60}_{0.48}$ | $\mathbf{69.84}_{1.06}$ | $\mathbf{86.26}_{0.00}$ | $\mathbf{74.11}_{0.68}$ | $\mathbf{70.55}_{0.88}$ | $\mathbf{81.07}_{0.00}$ | $56.07_{0.85}$ | $46.09_{1.06}$ | $\mathbf{65.30}_{0.01}$ |
| Δ IFPL | ↑ 2.72 | ↑ 3.89 | ↑ 1.11 | ↑ 2.39 | ↑ 1.07 | ↑ 3.4 | ↑ 18.14 | ↑ 2.81 | ↑ 12.84 |
| Δ GRIP | ↑ 7.64 | ↑ 4.17 | ↑ 5.29 | ↑ 5.89 | ↑ 7.48 | ↑ 8.96 | ↑ 18.97 | ↑ 1.13 | ↑ 19.00 |
| **Visual prompts** | | | | | | | | | |
| FPL | $67.03_{0.65}$ | $\mathbf{65.50}_{0.41}$ | $71.94_{0.00}$ | $65.14_{0.25}$ | $62.24_{0.22}$ | $67.85_{0.00}$ | $47.60_{1.09}$ | $47.69_{0.48}$ | $52.43_{0.00}$ |
| IFPL | $\mathbf{68.69}_{0.45}$ | $\mathbf{66.12}_{0.46}$ | $76.91_{0.00}$ | $67.11_{1.19}$ | $62.93_{1.23}$ | $73.53_{0.00}$ | $\mathbf{51.65}_{0.70}$ | $50.34_{0.65}$ | $57.86_{0.01}$ |
| GRIP | $67.95_{1.2}$ | $63.09_{0.56}$ | $\mathbf{77.18}_{0.00}$ | $\mathbf{71.22}_{0.77}$ | $\mathbf{68.43}_{0.61}$ | $\mathbf{82.19}_{0.00}$ | $54.57_{4.86}$ | $50.51_{0.99}$ | $\mathbf{62.78}_{0.00}$ |
| Δ IFPL | ↑ 1.66 | ↓ 0.38 | ↑ 4.97 | ↑ 1.97 | ↑ 0.69 | ↑ 5.68 | ↑ 4.05 | ↑ 2.65 | ↑ 5.43 |
| Δ GRIP | ↑ 0.92 | ↓ 3.41 | ↑ 5.24 | ↑ 6.08 | ↑ 6.19 | ↑ 14.34 | ↑ 6.97 | ↑ 2.82 | ↑ 10.35 |
| **Multimodal prompts** | | | | | | | | | |
| FPL | $72.54_{0.36}$ | $\mathbf{65.26}_{0.38}$ | $77.47_{0.00}$ | $62.84_{1.05}$ | $62.32_{0.65}$ | $71.43_{0.00}$ | $43.71_{2.19}$ | $44.85_{0.31}$ | $54.86_{0.00}$ |
| IFPL | $\mathbf{73.14}_{0.87}$ | $65.39_{1.33}$ | $81.47_{0.00}$ | $70.60_{1.04}$ | $63.69_{0.53}$ | $46.04_{0.36}$ | $\mathbf{53.21}_{1.24}$ | $\mathbf{47.59}_{1.04}$ | $43.17_{0.25}$ |
| GRIP | $\mathbf{74.56}_{2.02}$ | $64.82_{1.63}$ | $\mathbf{82.01}_{0.00}$ | $\mathbf{73.78}_{0.91}$ | $\mathbf{69.37}_{0.61}$ | $\mathbf{82.17}_{0.00}$ | $54.07_{2.25}$ | $47.37_{0.70}$ | $\mathbf{63.42}_{0.00}$ |
| Δ IFPL | ↑ 1.91 | ↑ 0.13 | ↑ 4.00 | ↑ 7.76 | ↑ 1.37 | ↓ 25.39 | ↑ 9.5 | ↑ 2.74 | ↓ 11.69 |
| Δ GRIP | ↑ 2.02 | ↓ 0.44 | ↑ 4.54 | ↑ 10.84 | ↑ 7.05 | ↑ 10.74 | ↑ 10.36 | ↑ 2.52 | ↑ 8.56 |

Table 2: For each learning paradigm, we compare FPL, IFPL, and GRIP on Flowers102, RESICS45, and DTD, for all the learning settings SSL, UL, TRZSL. We average across 5 runs and report the standard deviation. Δ METHOD is the difference between the accuracy of FPL and METHOD.

**Transductive zero-shot learning effectively transfers knowledge** In the TRZSL setting, GRIP improves over CLIP and the baselines by a large margin (Table 1). Figure 2 displays the balance of seen and unseen classes of each method alongside its accuracy. The *class balance* is $(acc_{unseen} - acc_{seen})/acc_{seen}$, where values close to zero indicate a good balance, negative values indicate better accuracies for seen classes, and positive values indicate better accuracies for unseen classes. Methods employing an iterative usage of pseudolabels maintain a good balance, as opposed to CoOp/VPT and FPL. This balance in accuracy is likely a combined effect of the quality of the pseudolabels and the transfer of knowledge from the seen to the unseen classes. The latter point is significant because it implies that even if we only possess unlabeled data for a specific target task, we can still use labeled data from related classes [30] within the same domain to enhance CLIP's performance.

**There is a trade-off between quality and quantity of pseudolabels** Table 2 shows the performance of CLIP employing prompts learned with different training strategies, all leveraging pseudolabels (Section 3.3). Iterative strategies are more effective than FPL which, similarly to [15], use a static set of a few pseudolabels for one iteration.

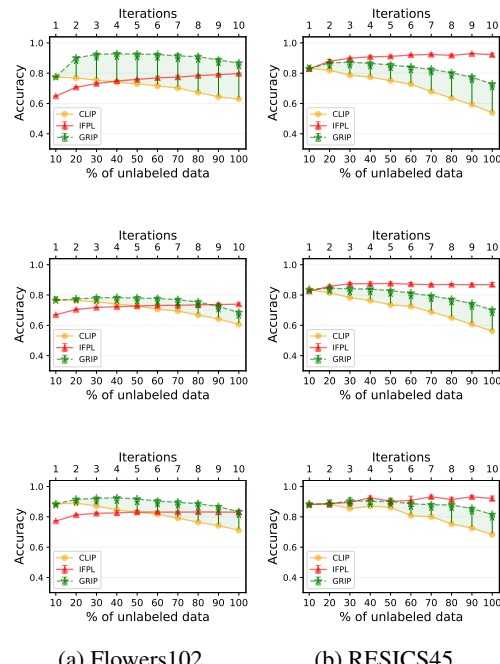

(a) Flowers102          (b) RESICS45

Figure 3: Evolution of pseudolabels accuracy during training. The rows refer to SSL, UL, and TRZSL, in order. IFPL refers to the top x-axis, while CLIP and GRIP to the bottom.

On Flowers102, RESICS45, and DTD, IFPL improves on average FPL by 5.6 points in SSL, 1.7 in UL, and 5.6 in TRZSL. GRIP boosts the performance even more by on average 7.8 points in SSL, 3.1 in UL, and 9.7 in TRZSL. Results on the other tasks are comparable or larger and we report them in Appendix A.3 due to space constraints.

Figure 3 shows the progression of pseudolabels quality for the iterative learning of textual prompts. IFPL maintains a fixed set of 16 pseudolabels, improving their quality with each iteration (top x-axis).

On the other hand, GRIP and CLIP expand pseudolabels by incorporating an additional decile of unlabeled data in each iteration (bottom x-axis). Initially, GRIP maintains accuracy, but as it nears completion the quality tends to decrease, while a larger dataset with good-quality pseudolabels becomes available.

Comparing GRIP and CLIP, GRIP's expanded pseudolabels exhibit superior quality and performs better (Table 1). Even though IFPL's pseudolabel accuracy surpasses GRIP in the final iteration, GRIP's overall performance remains better due to training on a larger number of pseudolabels (Table 2). This suggests that numerous, slightly noisier pseudolabels can yield better results, highlighting a trade-off and offering insights for future approaches.

**GRIP benefits adaptation even for larger image encoders** We measure how much the effect of the iterative strategies changes if we consider a larger pre-trained image encoder. In Table 3, we report the average improvements of GRIP on CLIP for Flowers102, RESICS45, and DTD. The magnitude of improvements slightly decreases when using a larger image encoder. However, we still see significant benefits for both modalities. The smaller relative improvements with respect to smaller visual encoders align with our expectations. Larger encoders possess a stronger base knowledge, making it relatively more challenging to attain further improvements on top of it. In Table 10, we break down the accuracy of CLIP with different backbones. The performance of the larger backbone is higher, indicating higher quality pseudolabels.

| **Textual prompts** | | | |
|---|---|---|---|
| | SSL | UL | TRZSL |
| Avg. $\Delta$ CLIP (ViT-B/32) | $17.46_{12.83}$ | $8.36_{2.85}$ | $23.77_{2.51}$ |
| Avg. $\Delta$ CLIP (ViT-L/14) | $15.85_{6.44}$ | $8.16_{6.12}$ | $19.96_{6.19}$ |
| **Visual prompts** | | | |
| | SSL | UL | TRZSL |
| Avg. $\Delta$ CLIP (ViT-B/32) | $10.78_{4.28}$ | $6.88_{-0.58}$ | $20.28_{13.78}$ |
| Avg. $\Delta$ CLIP (ViT-L/14) | $7.61_{3.27}$ | $4.89_{-0.48}$ | $16.14_{11.13}$ |

Table 3: Average improvement of GRIP with different backbones on Flowers102, RESICS45, and DTD. $\Delta$ CLIP is the difference between the accuracy of GRIP and CLIP. Alongside the average, we provide the minimum improvement across tasks.

## 4.2 The Robin Hood effect

Although training models with pseudolabels can lead to good performance, it can also result in biased predictions and generate disparate impacts on sub-populations, i.e., the "Matthew effect" [49, 8]. Particularly, the use of pseudolabels can lead to improved performance in *well-behaved* (high accuracy) classes but can cause stagnation or decreased performance in *poorly behaved* (low accuracy) classes. As we explore the use of pseudolabels, we investigate how the accuracy of the analyzed approaches distributes across classes. Figure 4 show an opposite scenario from typical SSL. The solid line represents the sorted per-class accuracies of CLIP. The arrows indicate the per-class accuracies of GRIP. For all learning paradigms, the iterative training strategies increase the accuracy of classes where CLIP is not proficient, while maintaining or decreasing the accuracy of initially well-behaved classes. This effect, which we call the "Robin Hood effect," is very interesting because it shows how CLIP can mitigate its own bias toward certain classes by learning from itself.

To understand the roots of the Robin Hood effect, we examine two factors: (1) the role of pseudolabels generated by CLIP, and (2) the role of prompt tuning. To disentangle these factors, we explore the variation in per-class accuracy of a basic linear classifier trained on CLIP's ViT-B/32 image representation.

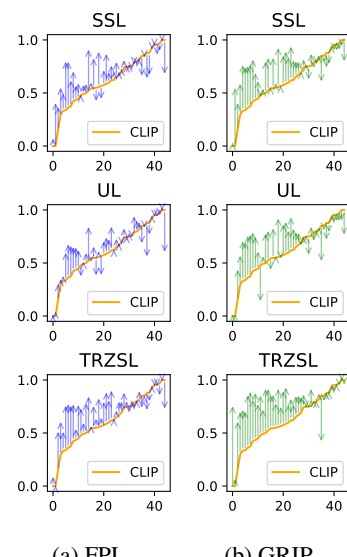

(a) FPL      (b) GRIP

Figure 4: Improvements of FPL and GRIP on CLIP's per-class accuracies (RESICS45). The x-axis is the ranked class index, while y-axis is the accuracy.

**"Second generation" pseudolabels are a good treatment for class disparity** We train the linear classifier in the SSL setting on 2 labeled examples per class and pseudolabels. The pseudolabels are obtained through conventional methods, where a threshold of .95 is applied, or by using CLIP to generate 16 pseudolabels per class.

We find that both approaches yield similar overall accuracies. However, we observe the "Matthew effect" when using the first approach. In contrast, when using CLIP-based pseudolabels, the class disparity of the regressor trained solely on seen classes is reduced. Particularly, we see a significant improvement on initially poor classes, together with a significant diminishing of the accuracy of well-behaved classes. We observe a clear manifestation of the "Robin Hood effect." We present plots illustrating this effect in Appendix A.4.

**Prompt tuning retains the accuracy of already rich classes better than linear probing**   To evaluate the role of prompt tuning in the "Robin Hood effect," we train a linear classifier and textual prompts in the UL setting using GRIP's training strategy. Comparing the per-class acurracies of the two approaches, GRIP on prompts shows an average improvement of 22.85 points for the poor classes across tasks, along with a slight average decrease of 0.3 points for the rich classes. On the other hand, linear probing determines a 14.42 points improvement for the poor classes, but it results in an average decrease of 9.39 points in accuracy for the rich classes (Appendix A.4).

## 5    Conclusions

We show that prompt tuning using pseudolabels generated by CLIP itself is a successful approach to enhance CLIP across various learning settings. Training strategies that iteratively refine pseudolabels turn out to be effective ways of leveraging pseudolabeled data. These approaches not only enhance CLIP's accuracy but also mitigate model biases toward certain classes. We hope this work lays a solid groundwork for reducing reliance on labeled data when adapting pre-trained vision-language models like CLIP to new tasks.

**Limitations**   The effectiveness of the training strategies examined in this paper depends on both the strategies themselves and the quality of pseudolabels. The latter is particularly crucial. If CLIP performs poorly on a task, we may struggle to obtain a reliable set of pseudolabels to begin with, potentially diminishing CLIP's performance. Despite this potential risk, we have not observed any relevant failure of GRIP, even in tasks where CLIP's initial accuracy is extremely low (such as FGVCAircraft). Also, the pseudolabeling strategy we adopt involves selecting $K$ pseudolabels per class, which can create a strong assumption about the distribution of the training data if we attempt to cover all unlabeled data. During the final iteration, it is as if we assume a uniform class balance.

Another important consideration is the efficiency of the explored methods. Repeating the training process multiple times brings impressive improvements at the cost of a non-negligible increase of computation time. At each iteration, we generate pseudolabels for the unlabeled data from scratch. While we parallelized the pseudolabeling procedure to cut some of the cost, reducing those for the iterative training presents more significant challenges. We decided to mainly focus on the analysis of qualitative and quantitative effects of pseudolabels in prompt tuning. Future research should address budget constraints and investigate optimal stopping criteria for the iterative process, considering the possibility of reaching a plateau or decreased pseudolabel quality after a certain point, to maximize efficiency while maintaining performance.

## Acknowledgments and Disclosure of Funding

We are thankful to our reviewers for the fruitful and insightful discussions that contributed to the refinement of the paper. We also thank Reza Esfandiarpoor and Zheng-Xin Yong for the comments on our drafts. This material is based on research sponsored by Defense Advanced Research Projects Agency (DARPA) and Air Force Research Laboratory (AFRL) under agreement number FA8750-19-2-1006. The U.S. Government is authorized to reproduce and distribute reprints for Governmental purposes notwithstanding any copyright notation thereon. The views and conclusions contained herein are those of the authors and should not be interpreted as necessarily representing the official policies or endorsements, either expressed or implied, of Defense Advanced Research Projects Agency (DARPA) and Air Force Research Laboratory (AFRL) or the U.S. Government. We gratefully acknowledge support from Google and Cisco. Disclosure: Stephen Bach is an advisor to Snorkel AI, a company that provides software and services for data-centric artificial intelligence.

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

# A Appendix

We include here extra information that supports the results presented in the main body of the paper.

**Reproducibility** We have provided the code to run the experiments as supplementary material for the submission. However, we plan to release it as an open repository upon acceptance.

## A.1 Trainable Prompts

**Text Prompt Tuning** The primary objective of text prompt tuning is to improve the alignment between the class token and the image features extracted by the image encoder. This is achieved by adding learnable vectors, i.e., *prefix*, before the CLASS token to create a contextualized representation. Specifically, the sequence

$$\boldsymbol{t} = [\text{V}]_1[\text{V}]_2 \ldots [\text{V}]_M[\text{CLASS}]$$

is fed into the textual encoder, where each vector $[\text{V}]_m$ $(m \in 1, \ldots, M)$ has the same dimension as word embeddings, and $M$ is a hyperparameter that determines the length of the prefix.

Context Optimization (CoOp) [48] was the first work to explore continuous prompts for VLMs. Follow-up works have experimented with different training strategies to enhance the generalizability of the learned prompts while preserving the core concept of continuous vector tuning [34, 12, 27, 46, 13, 37].

Tuning the text prefix vector changes the resulting $n$ linear weight vectors $w_i = \psi(p_i)$, while leaving the image features unchanged. Therefore, text prompt tuning may be most beneficial when image features are well-separated by class but may not be aligned with the corresponding textual prompt. Conversely, text prompt tuning may not be as effective when the image features are poorly separated, as in specialized or novel domains where CLIP may lack sufficient training data.

**Visual Prompt Tuning** Instead of tuning the text prompts, one can also tune the inputs of the vision encoder. In this case, a learnable visual prefix is prepended to the image tokens as input to the image transformer as follows:

$$\hat{\boldsymbol{I}} = [\text{p}]_1 \ldots [\text{p}]_K[\text{I}]_1 \ldots [\text{I}]_P$$

where $p$ represents a sequence of $K$ learnable prefix vectors, and $[\text{I}]_1 \ldots [\text{I}]_P$ are the image tokens from the corresponding $P$ patches of the input images. The new sequence $\hat{\boldsymbol{I}}$ is the input to the image encoder $\phi$.

Visual Prompt Tuning (VPT) was introduced in the context of efficiently adapting pre-trained vision transformers to downstream tasks [18]. However, the approach has since been applied in the context of VLM [34].

Whereas text prompt tuning does not alter the image features, visual prompt tuning does. By rearranging the image features within the projection space, VPT has the potential to improve CLIP when the image features are not well separated by class, such as in specialized domains.

**Multimodal Prompt Tuning** The previous approaches are unimodal, as they either involve modifying the text or visual input, but never both. This choice may be suboptimal as it does not allow the flexibility to dynamically adjust both representations on a downstream task. Recently, multimodal prompt tuning has been introduced [44, 19]. We focus on Unified Prompt Tuning (UPT) [44] which essentially learns a tiny neural network to jointly optimize prompts across different modalities. UPT learns a set of prompts $\boldsymbol{U} = [\boldsymbol{U}_T, \boldsymbol{U}_V] \in \mathbb{R}^{d \times n}$ with length $n$, where $\boldsymbol{U}_T \in \mathbb{R}^{d \times n_T}$, $\boldsymbol{U}_V \in \mathbb{R}^{d \times n_V}$. $\boldsymbol{U}$ is transformed as follows:

$$\boldsymbol{U}' = \text{SA}(\boldsymbol{U}) + \text{LN}(\boldsymbol{U})$$
$$\hat{\boldsymbol{U}} = \text{FFN}\left(\text{LN}\left(\boldsymbol{U}'\right)\right) + \text{LN}\left(\boldsymbol{U}'\right)$$

where SA is the self-attention operator, LN is the layer normalization operator, and FFN is a feed forward network. After transformation, we obtain $\hat{\boldsymbol{U}} = \left[\hat{\boldsymbol{U}_T}, \hat{\boldsymbol{U}_V}\right] \in \mathbb{R}^{d \times n}$, such that $\hat{\boldsymbol{U}_T}$ is to be used as a text prompt, and $\hat{\boldsymbol{U}_V}$ is to be used as a visual prompt.

| | Num. classes ($|\mathcal{Y}|$) | Num. seen classes ($|S|$) | Num. unseen classes ($|U|$) | Size training data | Avg. labeled data per class | Size test |
|---|---|---|---|---|---|---|
| Flowers102 | 102 | 63 | 39 | 2040 | 16 | 6149 |
| RESICS45 | 45 | 27 | 18 | 6300 | 110 | 25200 |
| FGVC-Aircraft | 100 | 62 | 38 | 6667 | 53 | 3333 |
| MNIST | 10 | 6 | 4 | 60000 | 4696 | 10000 |
| EuroSAT | 10 | 6 | 4 | 27000 | 2200 | 5000 |
| DTD | 47 | 29 | 18 | 3760 | 64 | 1880 |

Table 4: For each dataset we report the number of classes, the number of seen and unseen classes in the TRZSL setting, the size of training data (including both labeled and unlabeled data), the average number of labeled examples per class, and the size of the test set which is the same across learning paradigms. We recall that we use the datasets gathered by the recent ELEVATER [23] benchmark for vision-language models.

The author of UPL argue that self-attention allows for beneficial interaction between the two separate modalities, which leads to both separable visual features, and text classifiers that are well-aligned with the corresponding visual features [44].

**Prompts initialization** We initialize textual and visual prompts from a normal distribution of mean 0 and variance 0.02. We note that we learn shallow visual prompts by modifying only the input to the image encoder. Multimodal prompts are initialized from a uniform distribution. We found that the latter was not working properly for textual and visual prompts.

**Additional training settings** For training, the batch size is 64.

**Additional details about pseudolabels assignment** It can happen that if $K$ is too large and the unlabeled dataset is smaller than $K \times C$, we cannot assign $K$ samples per class. In this case, we can reduce $K$ accordingly. Also, we the same sample might get assigned to the pseudolabel lists of different classes. It rarely happens in our experiments cases. Hoever, this is a characteristic of the pseudolabeling strategy proposed in [15], and it can be an object of study for future work motivated by the effectiveness of self-training.

## A.2 Datasets details

We use six datasets from specialized or fine-grained domains. Here we provide a description of each of them. In Table 4, we report the details about the number of classes and data available for each dataset. For each dataset, we also show CLIP's prediction distribution over classes Figure 5.

**Flowers102 [29]** It is a dataset collecting images for 102 flower categories commonly occurring in the United Kingdom. For each class we have between 40 and 258 images. Figure 5a shows that CLIP's predictions are skewed toward certain classes, which are predicted more often than what we would expect according to the real class distribution on the test set.

**RESICS45 [9]** This is a publicly available benchmark for Remote Sensing Image Scene Classification. It collects 45 kind of scenes. Figure 5b shows that CLIP predicts more often a subset of classes.

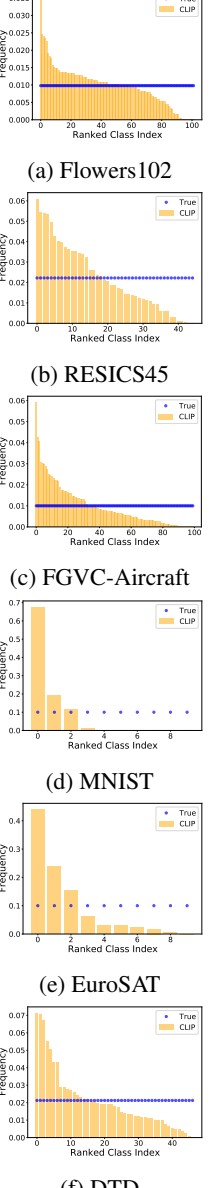

(a) Flowers102

(b) RESICS45

(c) FGVC-Aircraft

(d) MNIST

(e) EuroSAT

(f) DTD

Figure 5: For each dataset we show the distribution of CLIP's predictions over classes on the test set. The blue dots represent the true class distribution.

**Multimodal prompts**

| | Flowers102 | | | RESICS45 | | | FGVCAircraft | | |
|---|---|---|---|---|---|---|---|---|---|
| Method | SSL | UL | TRZSL | SSL | UL | TRZSL | SSL | UL | TRZSL |
| CLIP | $63.67_{0.00}$ | | $63.40_{0.00}$ | $54.48_{0.00}$ | | $54.46_{0.00}$ | $\mathbf{17.58_{0.00}}$ | | $\mathbf{17.86_{0.00}}$ |
| UPT | $68.03_{1.29}$ | - | $61.05_{0.04}$ | $62.84_{1.05}$ | - | $58.79_{0.04}$ | $11.13_{4.98}$ | - | $15.89_{0.07}$ |
| GRIP | $\mathbf{74.56_{2.02}}$ | $64.82_{1.63}$ | $\mathbf{82.01_{0.01}}$ | $\mathbf{73.68_{0.91}}$ | $69.37_{0.61}$ | $\mathbf{82.17_{0.00}}$ | $17.36_{0.43}$ | $14.73_{0.08}$ | $\mathbf{17.85_{10.30}}$ |
| Δ CLIP | ↑ 10.89 | ↑ 1.15 | ↑ 18.61 | ↑ 19.2 | ↑ 14.89 | ↑ 27.71 | ↓ 0.22 | ↓ 2.85 | ↓ 0.01 |
| Δ UPT | ↑ 6.53 | - | ↑ 20.96 | ↑ 10.84 | - | ↑ 22.38 | ↑ 6.23 | - | ↑ 1.96 |
| | MNIST | | | EuroSAT | | | DTD | | |
| CLIP | $25.10_{0.00}$ | | $20.77_{0.00}$ | $32.88_{0.00}$ | | $30.54_{0.00}$ | $43.24_{0.00}$ | | $43.45_{0.00}$ |
| UPT | $64.44_{3.66}$ | - | $63.59_{0.11}$ | $68.85_{9.92}$ | - | $60.43_{0.04}$ | $43.71_{2.18}$ | - | $36.91_{0.04}$ |
| GRIP | $65.94_{2.23}$ | $68.18_{run}$ | $73.75_{2.93}$ | $60.38_{4.77}$ | $61.52_{3.04}$ | $\mathbf{95.52_{0.40}}$ | $\mathbf{54.07_{2.25}}$ | $\mathbf{47.37_{0.7}}$ | $\mathbf{63.42_{0.00}}$ |
| Δ CLIP | ↑ 40.84 | ↑ 43.08 | ↑ 52.98 | ↑ 27.5 | ↑ 28.64 | ↑ 64.98 | ↑ 10.83 | ↑ 4.13 | ↑ 19.97 |
| Δ UPT | ↑ 2.35 | - | ↑ 10.16 | ↓ 8.47 | - | ↑ 35.09 | ↑ 10.36 | - | ↑ 26.51 |

Table 5: For each learning paradigm, we compare the accuracy of GRIP with CLIP zero-shot (ViT-B/32), and UPT. Results are for SSL, UL, and TRZSL on FRAMED. We average the accuracy on 5 seeds and report the standard deviation. Δ METHOD is the difference between the accuracy of GRIP and METHOD. We note that for UL we can not apply UPL since no labeled data is available.

**FGVC-Aircraft [26]**  It describes the fine-grained task of categorizing aircraft. We consider the task of classifying aircrafts into 100 variants. Also for this task, CLIP assigns images to a reduced set of classes (Figure 5c).

**MNIST [11]**  MNIST is a database of handwritten digits. The digits are size-normalized and centered in a fixed-size image. We observe that CLIP never predicts 6 out of 10 classes (Figure 5c).

**EuroSAT [14]**  EuroSAT represents the task of categorizing satellite images of scenes. It consists of 10 classes. In Figure 5e, we show CLIP's predictions distribution over the classes.

**DTD [10]**  DTD stands for Describable Textures Dataset. It is an evolving collection of textural images in the wild, and it is annotated relying on human-centric attributes, inspired by the perceptual properties of textures. The zero-shot CLIP predictions show the model's bias toward certain classes (Figure 5f).

## A.3  Experiments

In this section, we report tables and plots that complement the results presented in Section 4.

**The effect of GRIP on multimodal prompts**  Table 5 shows the improvements of GRIP on CLIP and Unified Prompt Tuning (UPL) [44]. Similar to the results in Table 1, GRIP consistently improves CLIP with respect to the baselines. The improvements on CLIP are by 18.2 in semi-supervised learning, 14.8 in unsupervised learning, and 30.7 in transductive zero-shot learning. While GRIP outperforms UPL by 4.7 in semi-supervised learning, and 19.5 in transductive zero-shot learning.

**Comparison across iterative strategies**  In Table 6, we report a comparison between FPL and the iterative strategies (IFPL and GRIP) on MNIST, EuroSAT, and FGVC-Aircraft. Results on the other tasks can be found in the main body of the paper Section 4.1. While GRIP largely and consistently outperforms FPL by on average 16.7 points in accuracy, IFPL is not robust and it leads to performances that are inferior to FPL by on average 4.4 points in accuracy.

**The evolving accuracy of dynamic pseudolabels**  Figure 6 represents the evolution of pseudolabels accuracy during training for all datasets, but Flowers102 and RESICS45 presented in Figure 3. We observe that the accuracy of the pseudolabels characterizes the overall performance of the models reported in Table 6. For instance, IFPL for EuroSAT in the TRZSL setting is highly variable, explaining the low average accuracy of the model on the test set (Table 6). Similarly, for MNIST in the TRZSL we observe that after the first iteration, the pseudolabels get very noisy. When we increase the amount of pseudolabeled data, the accuracy of CLIP is not necessarily constant. This is because as we increase K, we are effectively selecting pseudolabels with lower similarities to the

**Textual prompts**

| Method | MNIST | | | EuroSAT | | | FGVCAircraft | | |
|---|---|---|---|---|---|---|---|---|---|
| | SSL | UL | TRZSL | SSL | UL | TRZSL | SSL | UL | TRZSL |
| FPL | $66.06_{1.10}$ | $40.03_{2.63}$ | $9.73_{19.45}$ | $\mathbf{62.05}_{1.64}$ | $48.96_{1.49}$ | $53.70_{26.87}$ | $\mathbf{20.02}_{0.77}$ | $\mathbf{16.62}_{0.67}$ | $17.55_{0.37}$ |
| IFPL | $59.14_{3.43}$ | $28.94_{2.05}$ | $0.00_{0.00}$ | $61.28_{1.59}$ | $\mathbf{56.46}_{3.26}$ | $14.36_{28.71}$ | $18.00_{0.35}$ | $13.80_{0.67}$ | $21.72_{0.77}$ |
| GRIP | $\mathbf{71.78}_{3.59}$ | $\mathbf{67.88}_{2.76}$ | $\mathbf{74.06}_{0.29}$ | $58.66_{2.64}$ | $57.21_{1.77}$ | $\mathbf{92.33}_{0.69}$ | $16.98_{0.82}$ | $15.22_{0.71}$ | $\mathbf{26.08}_{0.25}$ |
| Δ IFPL | ↓ 6.92 | ↓ 11.09 | ↓ 9.73 | ↓ 0.77 | ↑ 7.50 | ↓ 39.34 | ↓ 2.02 | ↓ 2.82 | ↑ 4.17 |
| Δ GRIP | ↑ 5.72 | ↑ 27.85 | ↑ 64.33 | ↓ 3.39 | ↑ 8.25 | ↑ 38.63 | ↓ 3.04 | ↓ 1.40 | ↑ 8.53 |

**Visual prompts**

| Method | MNIST | | | EuroSAT | | | FGVCAircraft | | |
|---|---|---|---|---|---|---|---|---|---|
| | SSL | UL | TRZSL | SSL | UL | TRZSL | SSL | UL | TRZSL |
| FPL | $42.84_{16.80}$ | $39.62_{6.53}$ | $31.82_{17.53}$ | $52.47_{2.53}$ | $48.79_{3.69}$ | $68.68_{14.74}$ | $\mathbf{20.14}_{0.26}$ | $\mathbf{18.28}_{0.33}$ | $16.28_{0.45}$ |
| IFPL | $52.91_{8.99}$ | $37.17_{6.27}$ | $38.38_{4.21}$ | $57.85_{6.52}$ | $32.52_{10.00}$ | $48.13_{11.13}$ | $18.77_{0.48}$ | $16.36_{0.37}$ | $19.29_{0.36}$ |
| GRIP | $\mathbf{69.66}_{5.51}$ | $\mathbf{68.04}_{1.11}$ | $\mathbf{69.54}_{1.31}$ | $\mathbf{63.48}_{3.09}$ | $\mathbf{63.68}_{3.42}$ | $\mathbf{96.97}_{0.77}$ | $19.43_{0.50}$ | $17.51_{0.61}$ | $\mathbf{26.42}_{0.30}$ |
| Δ IFPL | ↑ 10.07 | ↓ 2.45 | ↑ 6.56 | ↑ 5.38 | ↓ 16.27 | ↓ 20.55 | ↓ 1.37 | ↓ 1.92 | ↑ 3.01 |
| Δ GRIP | ↑ 26.82 | ↑ 28.42 | ↑ 37.72 | ↑ 11.01 | ↑ 14.89 | ↑ 28.29 | ↓ 0.71 | ↓ 0.77 | ↑ 10.14 |

Table 6: For each learning paradigm, we compare FPL, IFPL, and GRIP on MNIST, EuroSAT, and FGVCAircraft. We average across 5 runs and report the standard deviation. Δ METHOD is the difference between the accuracy of FPL and METHOD.

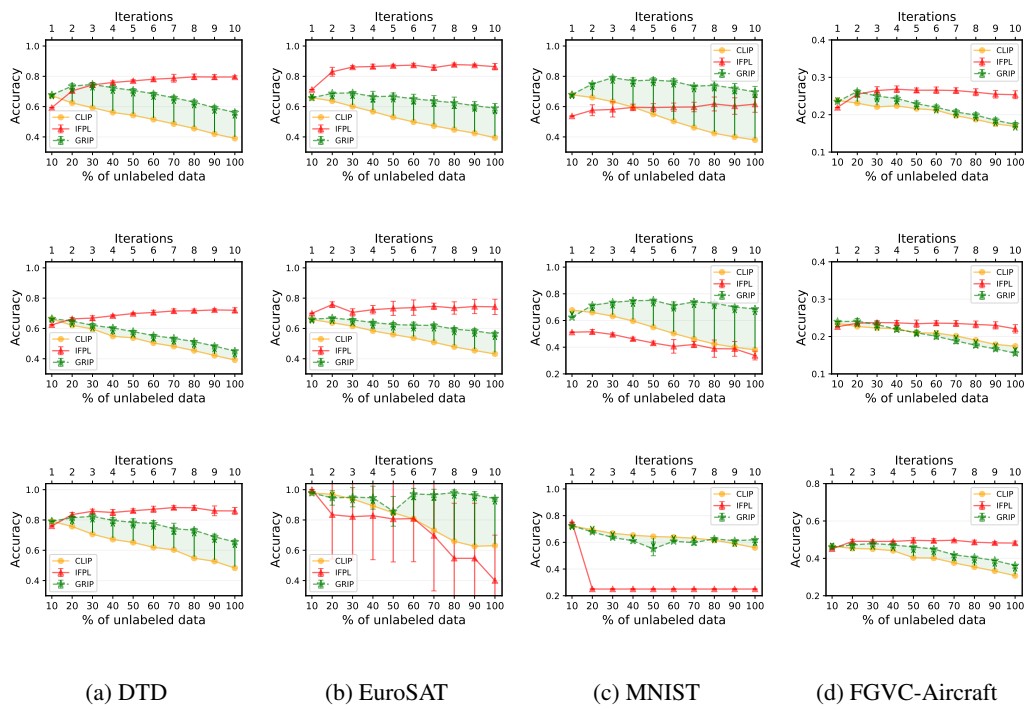

    (a) DTD         (b) EuroSAT         (c) MNIST         (d) FGVC-Aircraft

Figure 6: We plot the evolution of dynamic-pseudolabels accuracy during training. The rows refer to SSL, UL, and TRZSL, in order. IFPL refers to the top x-axis, while CLIP and GRIP to the bottom.

classes, resulting in a reduction in their accuracy, as shown in the plot. This observation aligns with previous findings in [15].

**GRIP performance on transductive zero-shot learning**    We show how the effectiveness of GRIP is consistent over the three random splits of seen and unseen classes which we randomly generated. The splits are reported in Table 9. Table 8 gathers the accuracy of seen and unseen classes, along with the harmonic mean for all three splits using textual prompts. Beyond the consistent improvement induced by GRIP training strategy, we observe that the accuracy of GRIP on the seen classes is often lower than the accuracy of CoOp on the same set of classes. During training, for large lambda, the loss component of unlabeled data (unseen classes) is the first to decrease, while the loss on the seen classes reduces later. Thus, we hypothesize that extra training steps might be needed to complete the learning on the labeled data.

## A.4  The Robin Hood effect

**The Robin Hood effect on all tasks**    For each dataset, we provide the per-class accuracy distribution of GRIP compared with CLIP, Figure 8. The Robin Hood effect characterizes all the tasks. We observe that for GRIP the increase in overall accuracy corresponds to consistent improvements in the predictions of initially poor classes. By comparing Figure 7 with Figure 8, we see that GRIP reinforces the Robin Hood effect already visible when using FPL in certain cases.

**The importance of good quality pseudolabels to mitigate the Matthew effect in SSL**    In the SSL setting, we train a logistic regression on top of the visual feature extracted by CLIP's image encoder (ViT-B/32). In Figure 9, we show the per-class accuracy of the final model trained by combining labeled data with either pseudolabels assigned with the conventional scheme (threshold at .95) or 16 CLIP-generated pseudolabels. We compare the two distribution with the per-class accuracy of the model trained solely on the few labeled examples per class (2 instances).

**The different impact of prompt tuning and linear probing on the Robin Hood effect**    We investigate if there is any difference in the Robin Hood effect when adapting CLIP via prompt tuning or linear probing. We train both relying on the iterative training strategy that grows the set of pseudolabels at each iteration by using the top-$K$ scheme (Section 3). We consider the UL setting.

Among the set of target classes, we distinguish between *poor* and *rich* classes. A class is *poor*, if CLIP's accuracy on that class is lower than its overall accuracy on the task. Otherwise, the class is considered *rich*. Table 7 reports the accuracy of the two approaches, and the accuracy on the poor and rich classes, while highlighting the average effect with respect to CLIP. Training with prompt tuning retains more knowledge of the rich classes than linear probing. Prompt tuning reduces the accuracy on the rich classes by on average 0.3 points, while linear probing has an average deterioration of 9.4. the reduction of accuracy of rich classes is on average 30 times larger than the reduction observed using prompts. Overall, GRIP works better than linear probing. We note that the lower accuracy of linear probing is characterized by a worse ability to correctly predict the rich classes, i.e., "rich get poorer." This is surprising, as we would have expected the errors to concentrate on the poor classes compared to CLIP.

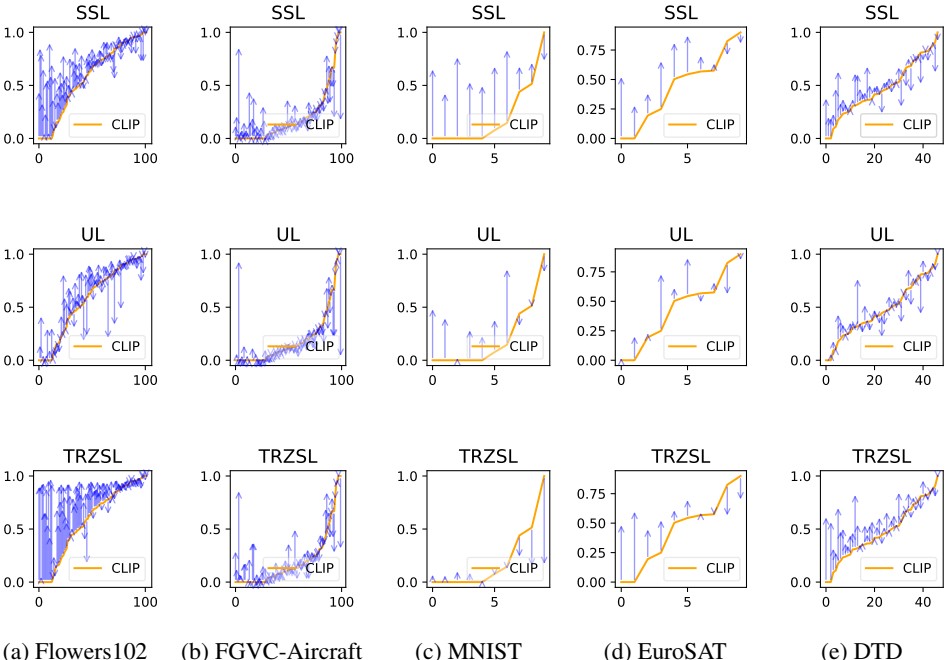

Figure 7: Per-class accuracy of FPL compared to CLIP's per-class accuracy on Flowers102, FGVC-Aircraft, MNIST, EuroSAT, DTD. **X-axis** is the ranked class index, while the **y-axis** is the accuracy.

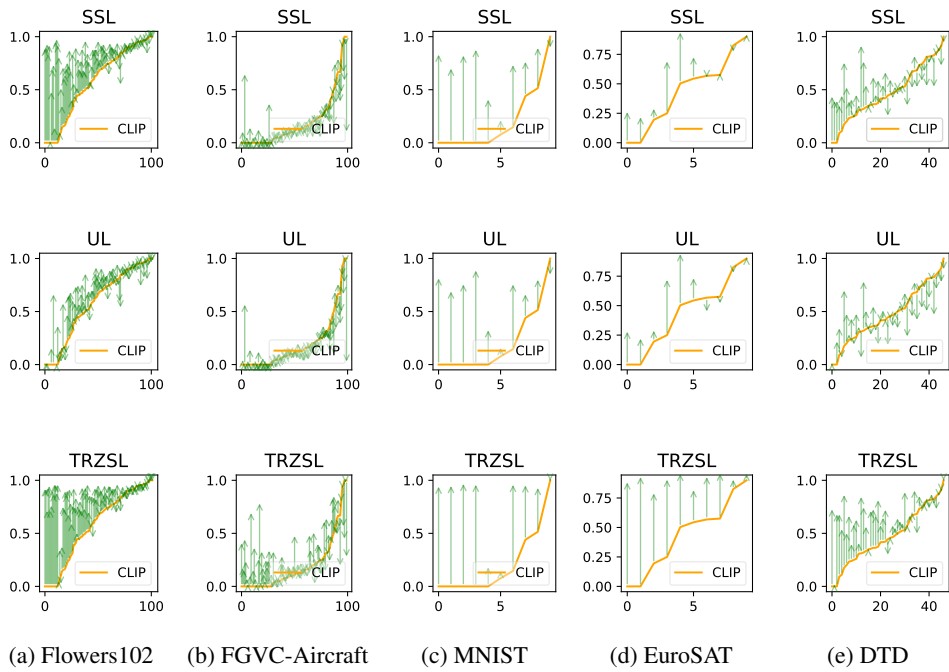

Figure 8: Per-class accuracy of GRIP compared to CLIP's per-class accuracy on Flowers102, FGVC-Aircraft, MNIST, EuroSAT, and DTD. **X-axis** is the ranked class index, while the **y-axis** is the accuracy.

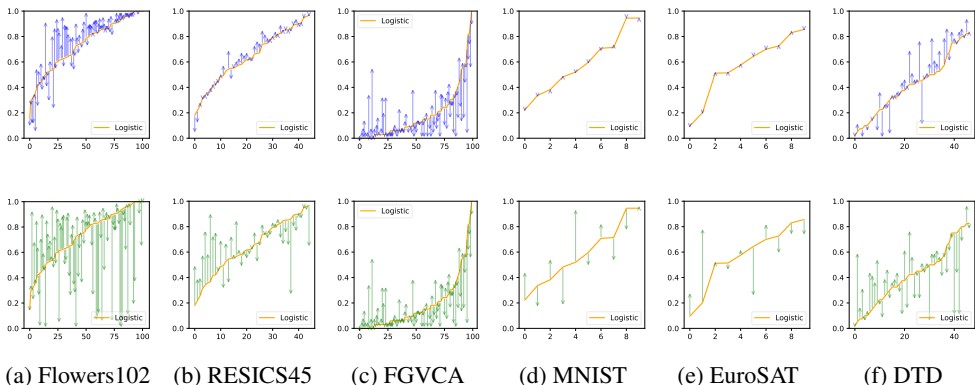

Figure 9: Per-class accuracy of a logistic classifier using conventional pseudolabels (first row) and CLIP-based pseudolabels (second row). The solid orange line represents the per-class accuracy of a logistic regression trained on 2-shots per class. **X-axis** is the ranked class index, while the **y-axis** is the accuracy. We present results for Flowers102, RESICS45, FGVC-Aircraft, MNIST, EuroSAT, and DTD, in order.

| | Flowers102 | RESICS45 | FGVC-Aircraft | MNIST | EuroSAT | DTD | Avg. Δ |
|---|---|---|---|---|---|---|---|
| Linear probe (LP) | 41.01 | 58.79 | 61.94 | 50.52 | 51.37 | 10.17 | - |
| GRIP | **46.09** | **70.55** | **69.84** | **57.21** | **67.88** | **15.22** | - |
| Rich CLIP | **67.81** | 75.47 | 85.16 | 65.26 | 65.14 | **45.93** | - |
| Rich LP | 52.87 | 69.01 | 79.55 | 67.53 | 50.34 | 29.12 | - |
| Rich GRIP | 56.05 | **78.81** | **86.40** | **71.73** | **77.84** | 31.95 | - |
| Δ LP | ↓ 14.92 | ↓ 6.47 | ↓ 5.61 | ↑ 2.26 | ↓ 14.79 | ↓ 16.81 | ↓ **9.39** |
| Δ GRIP | ↓ 11.76 | ↑ 3.33 | ↑ 1.24 | ↑ 6.46 | ↑ 12.70 | ↓ 13.98 | ↓ 0.33 |
| Poor CLIP | 25.63 | 35.60 | 27.98 | 11.10 | 3.18 | 5.35 | - |
| Poor LP | 26.50 | 42.77 | 36.25 | 28.34 | 56.76 | 4.77 | - |
| Poor GRIP | **35.03** | **56.85** | **42.82** | **39.88** | **65.08** | **6.31** | - |
| Δ LP | ↑ 0.87 | ↑ 7.18 | ↑ 8.27 | ↑ 17.24 | ↑ 53.58 | ↓ 0.58 | ↑ 14.43 |
| Δ GRIP | ↑ 9.4 | ↑ 21.26 | ↑ 14.84 | ↑ 28.78 | ↑ 61.9 | ↑ 0.96 | ↑ **22.86** |

Table 7: For each task we report the overall accuracy of linear probing (LP) and GRIP textual along with the accuracy on *poor* and *rich* classes. Δ METHOD is the difference between the accuracy of CLIP and METHOD. For an overall evaluation of the difference between linear probing and prompt tuning, we report the average difference of LP and GRIP with respect to CLIP on poor and rich classes.

**Split 1**

| Method | Flowers102 | | | RESICS45 | | | FGVCAircraft | | |
|---|---|---|---|---|---|---|---|---|---|
| | S | U | H | S | U | H | S | U | H |
| CLIP | $64.26_{0.00}$ | $62.56_{0.00}$ | $63.4_{0.00}$ | $54.85_{0.00}$ | $54.08_{0.00}$ | $54.46_{0.00}$ | $16.27_{0.00}$ | $19.79_{0.00}$ | $17.86_{0.00}$ |
| CoOp | $\mathbf{91.52}_{0.36}$ | $48.35_{2.96}$ | $63.22_{2.60}$ | $\mathbf{84.66}_{1.01}$ | $50.73_{3.28}$ | $63.37_{2.23}$ | $\mathbf{34.18}_{1.56}$ | $16.28_{3.69}$ | $21.70_{3.45}$ |
| GRIP | $90.31_{0.51}$ | $\mathbf{82.57}_{1.26}$ | $\mathbf{86.26}_{0.81}$ | $82.68_{0.47}$ | $\mathbf{79.53}_{0.72}$ | $\mathbf{81.07}_{0.37}$ | $22.25_{0.07}$ | $\mathbf{31.51}_{0.59}$ | $\mathbf{26.08}_{0.25}$ |
| Δ CLIP | ↑ 26.05 | ↑ 20.01 | ↑ 22.86 | ↑ 27.83 | ↑ 25.45 | ↑ 26.61 | ↑ 5.98 | ↑ 11.72 | ↑ 8.22 |
| Δ CoOp | ↓ 1.21 | ↑ 34.22 | ↑ 23.04 | ↓ 1.98 | ↑ 28.8 | ↑ 17.7 | ↓ 11.93 | ↑ 15.23 | ↑ 4.38 |

| Method | MNIST | | | EuroSAT | | | DTD | | |
|---|---|---|---|---|---|---|---|---|---|
| | S | U | H | S | U | H | S | U | H |
| CLIP | $31.74_{0.00}$ | $15.43_{0.00}$ | $20.77_{0.00}$ | $22.33_{0.00}$ | $48.3_{0.00}$ | $30.54_{0.00}$ | $42.5_{0.00}$ | $44.44_{0.00}$ | $43.45_{0.00}$ |
| CoOp | $94.68_{5.64}$ | $15.43_{7.75}$ | $21.15_{12.18}$ | $82.91_{8.81}$ | $46.02_{9.23}$ | $58.64_{5.86}$ | $\mathbf{69.67}_{1.17}$ | $34.81_{3.44}$ | $46.3_{2.92}$ |
| GRIP | $\mathbf{95.13}_{0.11}$ | $\mathbf{60.63}_{0.44}$ | $\mathbf{74.06}_{0.29}$ | $\mathbf{91.75}_{0.53}$ | $\mathbf{92.91}_{0.91}$ | $\mathbf{92.33}_{0.70}$ | $68.26_{0.69}$ | $\mathbf{62.61}_{1.87}$ | $\mathbf{65.30}_{1.03}$ |
| Δ CLIP | ↑ 63.39 | ↑ 45.2 | ↑ 53.29 | ↑ 69.42 | ↑ 44.61 | ↑ 61.79 | ↑ 25.76 | ↑ 18.17 | ↑ 21.85 |
| Δ CoOp | ↑ 0.45 | ↑ 45.2 | ↑ 52.91 | ↑ 8.84 | ↑ 46.89 | ↑ 33.69 | ↓ 1.41 | ↑ 27.8 | ↑ 19.00 |

**Split 2**

| Method | Flowers102 | | | RESICS45 | | | FGVCAircraft | | |
|---|---|---|---|---|---|---|---|---|---|
| | S | U | H | S | U | H | S | U | H |
| CLIP | $65.38_{0.00}$ | $60.64_{0.00}$ | $62.92_{0.00}$ | $59.5_{0.00}$ | $47.06_{0.00}$ | $52.55_{0.00}$ | $17.30_{0.00}$ | $18.12_{0.00}$ | $17.70_{0.00}$ |
| CoOp | $\mathbf{91.8}_{1.32}$ | $47.75_{3.86}$ | $62.77_{3.31}$ | $\mathbf{86.54}_{1.92}$ | $48.00_{3.01}$ | $61.70_{2.17}$ | $\mathbf{33.59}_{4.12}$ | $19.57_{1.37}$ | $\mathbf{24.63}_{0.63}$ |
| GRIP | $88.84_{0.75}$ | $\mathbf{70.93}_{2.08}$ | $\mathbf{78.86}_{1.26}$ | $84.47_{0.41}$ | $\mathbf{84.09}_{1.01}$ | $\mathbf{84.28}_{0.73}$ | $22.13_{0.24}$ | $\mathbf{28.32}_{0.33}$ | $\mathbf{24.84}_{0.05}$ |
| Δ CLIP | ↑ 23.46 | ↑ 10.29 | ↑ 15.94 | ↑ 27.83 | ↑ 25.45 | ↑ 26.61 | ↑ 4.83 | ↑ 10.20 | ↑ 7.14 |
| Δ CoOp | ↓ 2.96 | ↑ 23.18 | ↑ 16.09 | ↓ 2.07 | ↑ 36.09 | ↑ 22.58 | ↓ 11.46 | ↑ 8.75 | ↑ 0.21 |

| Method | MNIST | | | EuroSAT | | | DTD | | |
|---|---|---|---|---|---|---|---|---|---|
| | S | U | H | S | U | H | S | U | H |
| CLIP | $15.99_{0.00}$ | $39.18_{0.00}$ | $22.71_{0.00}$ | $32.47_{0.00}$ | $33.1_{0.00}$ | $32.78_{0.00}$ | $45.43_{0.00}$ | $39.72_{0.00}$ | $42.39_{0.00}$ |
| CoOp | $90.6_{13.02}$ | $18.77_{9.12}$ | $30.29_{12.38}$ | $86.43_{3.23}$ | $47.16_{11.17}$ | $60.53_{8.42}$ | $\mathbf{70.4}_{1.99}$ | $32.53_{4.58}$ | $44.42_{4.63}$ |
| GRIP | $\mathbf{95.71}$ | $\mathbf{97.50}$ | $\mathbf{96.59}$ | $\mathbf{91.08}_{0.02}$ | $\mathbf{92.02}_{0.98}$ | $\mathbf{91.55}_{0.47}$ | $66.69_{0.53}$ | $\mathbf{56.19}_{1.18}$ | $\mathbf{60.99}_{0.69}$ |
| Δ CLIP | ↑ 85.12 | ↑ 50.76 | ↑ 79.32 | ↑ 58.61 | ↑ 58.92 | ↑ 58.77 | ↑ 21.26 | ↑ 16.47 | ↑ 18.6 |
| Δ CoOp | ↑ 6.11 | ↑ 71.20 | ↑ 57.19 | ↑ 4.65 | ↑ 44.86 | ↑ 31.02 | ↓ 3.71 | ↑ 23.66 | ↑ 16.57 |

**Split 3**

| Method | Flowers102 | | | RESICS45 | | | FGVCAircraft | | |
|---|---|---|---|---|---|---|---|---|---|
| | S | U | H | S | U | H | S | U | H |
| CLIP | $68.29_{0.00}$ | $57.25_{0.00}$ | $62.28_{0.00}$ | $56.02_{0.00}$ | $52.32_{0.00}$ | $54.10_{0.00}$ | $17.55_{0.00}$ | $17.71_{0.00}$ | $17.63_{0.00}$ |
| CoOp | $\mathbf{91.52}_{0.35}$ | $48.35_{2.95}$ | $63.22_{2.60}$ | $\mathbf{87.61}_{2.17}$ | $43.64_{4.97}$ | $58.14_{4.12}$ | $\mathbf{37.77}_{1.92}$ | $16.46_{3.23}$ | $22.77_{3.09}$ |
| GRIP | $90.09_{0.53}$ | $\mathbf{69.00}_{2.44}$ | $\mathbf{78.13}_{1.71}$ | $85.19_{0.15}$ | $\mathbf{75.58}_{3.17}$ | $\mathbf{80.07}_{1.79}$ | $22.07_{0.23}$ | $\mathbf{28.72}_{0.76}$ | $\mathbf{24.95}_{0.20}$ |
| Δ CLIP | ↑ 21.8 | ↑ 11.75 | ↑ 15.85 | ↑ 29.17 | ↑ 23.26 | ↑ 25.97 | ↑ 4.52 | ↑ 11.01 | ↑ 7.32 |
| Δ CoOp | ↓ 1.43 | ↑ 20.65 | ↑ 14.91 | ↓ 2.42 | ↑ 31.94 | ↑ 21.93 | ↓ 15.70 | ↑ 12.26 | ↑ 2.18 |

| Method | MNIST | | | EuroSAT | | | DTD | | |
|---|---|---|---|---|---|---|---|---|---|
| | S | U | H | S | U | H | S | U | H |
| CLIP | $10.59_{0.00}$ | $46.74_{0.00}$ | $17.27_{0.00}$ | $41.47_{0.00}$ | $19.60_{0.00}$ | $26.62_{0.00}$ | $45.52_{0.00}$ | $39.58_{0.00}$ | $42.34_{0.00}$ |
| CoOp | $89.6_{8.08}$ | $26.3_{12.88}$ | $39.4_{16.61}$ | $79.33_{9.37}$ | $43.38_{12.49}$ | $55.06_{8.62}$ | $\mathbf{70.53}_{3.11}$ | $24.94_{5.37}$ | $36.63_{5.57}$ |
| GRIP | $\mathbf{95.8}$ | $\mathbf{96.06}$ | $\mathbf{95.93}$ | $\mathbf{90.57}_{0.13}$ | $\mathbf{94.25}_{1.10}$ | $\mathbf{92.37}_{0.60}$ | $67.28_{0.74}$ | $\mathbf{58.94}_{2.78}$ | $\mathbf{62.81}_{1.75}$ |
| Δ CLIP | ↑ 79.81 | ↑ 56.88 | ↑ 73.22 | ↑ 49.1 | ↑ 74.65 | ↑ 65.75 | ↑ 21.76 | ↑ 19.36 | ↑ 20.47 |
| Δ CoOp | ↑ 5.20 | ↑ 77.29 | ↑ 65.64 | ↑ 11.24 | ↑ 50.87 | ↑ 37.31 | ↓ 3.25 | ↑ 34.00 | ↑ 26.18 |

Table 8: In the TRZSL settings, for each dataset and split, we compare the accuracy of GRIP textual with CLIP zero-shot (ViT-B/32), and CoOp. Results show the accuracy on seen ($S$) and unseen classes ($U$), and the harmonic mean ($H$). We average the accuracy on 5 seeds and report the standard deviation. Δ METHOD is the difference between the accuracy of GRIP and METHOD.

**Split 1**

| | Seen classes ($S$) | Unseen classes ($U$) |
|---|---|---|
| Flowers102 | canna lily, petunia, silverbush, prince of wales feathers, pincushion flower, bird of paradise, frangipani, hard-leaved pocket orchid, bearded iris, passion flower, tiger lily, lenten rose, cape flower, air plant, mexican petunia, common dandelion, magnolia, foxglove, hibiscus, camellia, orange dahlia, clematis, anthurium, bougainvillea, ruby-lipped cattleya, stemless gentian, oxeye daisy, spring crocus, king protea, cyclamen, fritillary, californian poppy, wild pansy, desert-rose, sunflower, rose, grape hyacinth, pink primrose, red ginger, corn poppy, watercress, colt's foot, blanket flower, monkshood, morning glory, siam tulip, barbeton daisy, bolero deep blue, carnation, tree poppy, globe thistle, english marigold, primula, wallflower, blackberry lily, fire lily, love in the mist, moon orchid, sweet pea, mallow, pelargonium, mexican aster, poinsettia | canterbury bells, snapdragon, spear thistle, yellow iris, globe flower, purple coneflower, peruvian lily, balloon flower, giant white arum lily, artichoke, sweet william, garden phlox, alpine sea holly, great masterwort, daffodil, sword lily, marigold, buttercup, bishop of llandaff, gaura, geranium, pink and yellow dahlia, cautleya spicata, japanese anemone, black-eyed susan, osteospermum, windflower, gazania, azalea, water lily, thorn apple, lotus, toad lily, columbine, tree mallow, hippeastrum, bee balm, bromelia, trumpet creeper |
| RESICS45 | beach, palace, roundabout, railway station, railway, thermal power station, river, airplane, island, bridge, basketball court, desert, runway, ground track field, sea ice, sparse residential, cloud, dense residential, wetland, mountain, meadow, baseball diamond, parking lot, storage tank, tennis court, commercial area, mobile home park | airport, ship, snowberg, chaparral, church, circular farmland, stadium, terrace, forest, freeway, golf course, harbor, industrial area, intersection, lake, medium residential, overpass, rectangular farmland |
| FGVC-Aircraft | Tu-134, Spitfire, Challenger 600, 737-700, F-A-18, E-170, 727-200, A300B4, Falcon 2000, DR-400, MD-87, CRJ-700 ERJ 145, Falcon 900, MD-80, DC-10, Il-76, Global Express, Gulfstream IV, Saab 340, Yak-42, CRJ-900, L-1011, A330-200, A321, 747-300, DC-3, A310, ATR-42, CRJ-200, Hawk T1, Fokker 100, ATR-72, PA-28, A319, 707-320, A318, A320, BAE-125, 747-200, ERJ 135, 737-800, SR-20, BAE 146-300, Beechcraft 1900, Cessna 172, A340-300, EMB-120, 737-900, 737-400, Cessna 208, MD-90, 777-300, A340-600, 737-600, 737-300, DHC-1, DC-6, A380, C-47, 767-200, BAE 146-200 | 737-200, 747-100, 747-400, 757-200, 757-300, 767-300, 767-400, 777-200, A330-300, A340-200, A340-500, An-12, Boeing 717, C-130, Cessna 525, Cessna 560, DC-8, DC-9-30, DH-82, DHC-6, DHC-8-100, DHC-8-300, Dornier 328, E-190, E-195, Embraer Legacy 600, Eurofighter Typhoon, F-16A-B, Fokker 50, Fokker 70, Gulfstream V, MD-11, Metroliner, Model B200, Saab 2000, Tornado, Tu-154 |
| MNIST | 4, 2, 9, 3, 0, 5 | 8, 1, 6, 7 |
| EuroSAT | industrial buildings or commercial buildings, brushland or shrubland, lake or sea, highway or road, annual crop land, pasture land | river, forest, permanent crop land, residential buildings or homes or apartments |
| DTD | knitted, pitted, studded, bumpy, spiralled, scaly, polka-dotted, veined, wrinkled, banded, flecked, stained, chequered, sprinkled, bubbly, grid, lined, crystalline, fibrous, meshed, zigzagged, pleated, braided, perforated, potholed, waffled, dotted, matted, gauzy | blotchy, smeared, cobwebbed, cracked, crosshatched, stratified, striped, swirly, woven, freckled, frilly, grooved, honeycombed, interlaced, lacelike, marbled, paisley, porous |

**Split 2**

| | Seen classes ($S$) | Unseen classes ($U$) |
|---|---|---|
| Flowers102 | prince of wales feathers, air plant, canterbury bells, bishop of llandaff, bee balm, desert-rose, purple coneflower, spring crocus, pelargonium, windflower, sunflower, bougainvillea, rose, spear thistle, bird of paradise, carnation, fritillary, grape hyacinth, mexican aster, monkshood, poinsettia, black-eyed susan, sweet pea, anthurium, wallflower, oxeye daisy, moon orchid, blackberry lily, hibiscus, frangipani , cautleya spicata, camellia, canna lily, passion flower, wild pansy, stemless gentian, balloon flower, gaura, thorn apple, morning glory, hard-leaved pocket orchid, japanese anemone, sword lily, daffodil, english marigold, globe flower, peruvian lily, barbeton daisy, siam tulip, tiger lily, foxglove, pink and yellow dahlia, pink primrose, alpine sea holly, artichoke, petunia, colt's foot, ruby-lipped cattleya, red ginger, primula, snapdragon, garden phlox, mexican petunia | globe thistle, king protea, yellow iris, giant white arum lily, fire lily, pincushion flower, corn poppy, sweet william, love in the mist, cape flower, great masterwort, lenten rose, bolero deep blue, marigold, buttercup, common dandelion, geranium, orange dahlia, silverbush, californian poppy, osteospermum, bearded iris, tree poppy, gazania, azalea, water lily, lotus, toad lily, clematis, columbine, tree mallow , magnolia, cyclamen, watercress, hippeastrum, mallow, bromelia, blanket flower, trumpet creeper |
| RESICS45 | railway station, snowberg, palace, beach, commercial area, mountain, parking lot, dense residential, sparse residential, rectangular farmland, railway, island, tennis court, baseball diamond, thermal power station, industrial area, golf course, meadow, ground track field, storage tank, circular farmland, forest, bridge, harbor, river, freeway, sea ice | airplane, airport, roundabout, basketball court, runway, ship, chaparral, church, stadium, cloud, terrace, desert, wetland, intersection, lake, medium residential, mobile home park, overpass |
| FGVC-Aircraft | A321, MD-80, 737-200, DC-8, Falcon 900, Saab 340, 767-200, F-A-18, DC-6, SR-20, DC-3, Saab 2000, Fokker 70, 747-400, 737-700, A340-300, A310, A319, A380, 737-800, C-47, Dornier 328, 737-300, Eurofighter Typhoon, Cessna 208, Challenger 600, 737-600, Yak-42, Hawk T1, Fokker 100, DHC-8-100, Gulfstream IV, Model B200, Embraer Legacy 600, CRJ-900, A330-200, 767-400, DC-9-30, DR-400, Falcon 2000, 727-200, DHC-8-300, C-130, Boeing 717, 737-400, 757-300, 767-300, Beechcraft 1900, BAE 146-300, 737-500, PA-28, DHC-6, 707-320, An-12, A330-300, CRJ-700, 747-200, ATR-42, A318, DC-10, 747-100, A340-500 | 737-900, 747-300, 757-200, 777-200, 777-300, A300B4, A320, A340-200, A340-600, ATR-72, BAE 146-200, BAE-125, Cessna 172, Cessna 525, Cessna 560, CRJ-200, DH-82, DHC-1, E-170, E-190, E-195, EMB-120, ERJ 135, ERJ 145, F-16A-B, Fokker 50, Global Express, Gulfstream V, Il-76, L-1011, MD-11, MD-87, MD-90, Metroliner, Spitfire, Tornado, Tu-134, Tu-154 |
| MNIST | 2, 8, 4, 9, 1, 6 | 0, 3, 5, 7 |
| EuroSAT | brushland or shrubland, river, industrial buildings or commercial buildings, lake or sea, forest, permanent crop land | annual crop land, highway or road, pasture land, residential buildings or homes or apartments |
| DTD | pitted, scaly, polka-dotted, bumpy, honeycombed, fibrous, veined, porous, lined, dotted, perforated, potholed, pleated, waffled, braided, wrinkled, paisley, gauzy, meshed, grid, studded, knitted, swirly, crosshatched, freckled, chequered, grooved, smeared, frilly | banded, blotchy, bubbly, spiralled, sprinkled, cobwebbed, cracked, stained, crystalline, stratified, striped, flecked, woven, zigzagged, interlaced, lacelike, marbled, matted |

**Split 3**

| | Seen classes ($S$) | Unseen classes ($U$) |
|---|---|---|
| Flowers102 | oxeye daisy, canterbury bells, clematis, siam tulip, cape flower, black-eyed susan, air plant, californian poppy, globe thistle, giant white arum lily, cyclamen, snapdragon, frangipani, buttercup, common dandelion, hippeastrum, columbine, spring crocus, bolero deep blue, spear thistle, barbeton daisy, poinsettia, peruvian lily, alpine sea holly, artichoke, sunflower, tiger lily, toad lily, magnolia, lenten rose, great masterwort, camellia, mallow, morning glory, lotus, sweet william, thorn apple, carnation, daffodil, corn poppy, cautleya spicata, marigold, hibiscus, tree poppy, balloon flower, osteospermum, english marigold, king protea, azalea, foxglove, watercress, blackberry lily, bearded iris, monkshood, mexican aster, orange dahlia, water lily, mexican petunia, sweet pea, pink primrose, primula, silverbush, pincushion flower | hard-leaved pocket orchid, moon orchid, bird of paradise , colt's foot, yellow iris, globe flower, purple coneflower, fire lily, fritillary, red ginger, grape hyacinth, prince of wales feathers, stemless gentian, garden phlox, love in the mist, ruby-lipped cattleya, sword lily, wallflower, petunia, wild pansy, pelargonium, bishop of llandaff, gaura, geranium, pink and yellow dahlia, japanese anemone, windflower, gazania, rose, passion flower, anthurium, desert-rose, tree mallow, canna lily, bee balm, bougainvillea, bromelia, blanket flower, trumpet creeper |
| RESICS45 | railway, parking lot, wetland, meadow, harbor, island, mobile home park, storage tank, industrial area, bridge, baseball diamond, sea ice, runway, airplane, thermal power station, circular farmland, basketball court, roundabout, commercial area, railway station, terrace, forest, rectangular farmland, lake, medium residential, snowberg, river | airport, beach, ship, chaparral, church, sparse residential, cloud, stadium, dense residential, desert, tennis court, freeway, golf course, ground track field, intersection, mountain, overpass, palace |
| FGVC-Aircraft | An-12, 737-200, F-16A-B, BAE 146-200, MD-80, E-170, Gulfstream IV, DR-400, 737-900, 777-200, Boeing 717, 747-100, Saab 340, Cessna 525, Challenger 600, MD-90, DHC-8-100, Cessna 172, C-47, 747-400, BAE-125, 767-300, Cessna 560, A330-300, E-195, 737-500, Fokker 50, ATR-72, BAE 146-300, Fokker 70, Falcon 900, Falcon 2000, Spitfire, A340-200, DC-3, A340-300, Beechcraft 1900, A320, Hawk T1, E-190, Gulfstream V, Tu-134, 767-400, CRJ-200, 737-400, 747-300, Eurofighter Typhoon, PA-28, MD-87, Yak-42, DHC-1, 737-800, A380, Model B200, ERJ 135, SR-20, 737-300, 707-320, DC-10, Dornier 328, A300B4 | 727-200, 737-600, 737-700, 747-200, 757-200, 757-300, 767-200, 777-300, A310, A318, A319, A321, A330-200, A340-500, A340-600, ATR-42, C-130, Cessna 208, CRJ-700, CRJ-900, DC-6, DC-8, DC-9-30, DH-82, DHC-6, DHC-8-300, EMB-120, Embraer Legacy 600, ERJ 145, F-A-18, Fokker 100, Global Express, Il-76, L-1011, Metroliner, Saab 2000, Tornado, Tu-154 |
| MNIST | 8, 3, 5, 6, 1, 7 | 0, 9, 2, 4 |
| EuroSAT | river, highway or road, pasture land, permanent crop land, forest, residential buildings or homes or apartments | annual crop land, lake or sea, brushland or shrubland, industrial buildings or commercial buildings |
| DTD | pitted, pleated, polka-dotted, sprinkled, grooved, knitted, matted, wrinkled, honeycombed, chequered, braided, zigzagged, spiralled, banded, waffled, crosshatched, bubbly, smeared, dotted, porous, woven, freckled, lined, potholed, lacelike, marbled, stratified, scaly, studded | blotchy, bumpy, stained, cobwebbed, cracked, striped, crystalline, swirly, fibrous, flecked, veined, frilly, gauzy, grid, interlaced, meshed, paisley, perforated |

Table 9: For each dataset, we report the class names of seen and unseen classes in each of the splits used for TRZSL.

**Textual prompts**

| Method | Flowers102 | | | RESICS45 | | | DTD | | |
|---|---|---|---|---|---|---|---|---|---|
| | SSL | UL | TRZSL | SSL | UL | TRZSL | SSL | UL | TRZSL |
| CLIP (ViT-B/32) | 63.67 | 63.67 | 63.40 | 54.48 | 54.48 | 54.46 | 43.24 | 43.24 | 43.45 |
| GRIP (ViT-B/32) | 83.60 | 69.84 | 86.26 | 74.11 | 70.55 | 81.07 | 56.07 | 46.09 | **65.30** |
| CLIP (ViT-L/14) | 73.98 | 73.98 | 73.05 | 62.67 | 62.67 | 62.13 | 52.45 | 52.45 | 51.61 |
| GRIP (ViT-L/14) | **94.21** | **82.33** | **96.18** | **81.53** | **76.86** | **86.88** | **60.91** | **54.40** | 64.92 |

**Visual prompts**

| Method | Flowers102 | | | RESICS45 | | | DTD | | |
|---|---|---|---|---|---|---|---|---|---|
| CLIP (ViT-B/32) | 63.67 | 63.67 | 63.40 | 54.48 | 54.48 | 54.46 | 43.24 | 43.24 | 43.45 |
| GRIP (ViT-B/32) | 67.95 | 63.09 | 77.18 | 71.22 | 68.43 | 82.19 | 54.57 | 50.51 | **62.78** |
| CLIP (ViT-L/14) | 73.98 | **73.98** | 73.05 | 62.67 | 62.67 | 62.13 | 52.45 | 52.45 | 51.61 |
| GRIP (ViT-L/14) | **78.68** | 73.50 | **85.85** | **77.53** | **76.00** | **86.63** | **55.72** | **54.27** | 62.74 |

Table 10: Performance of CLIP and GRIP with different backbones on Flowers102, RESICS45, and DTD, for all the learning settings SSL, UL, TRZSL.

