# OpenReview forum: "Enhancing CLIP with CLIP: Exploring Pseudolabeling for Limited-Label Prompt Tuning"
_NeurIPS.cc/2023/Conference — NeurIPS 2023 poster_

### Official Review · Reviewer_qyTk · 2023-07-03

**Soundness:** 2 fair
**Presentation:** 3 good
**Contribution:** 2 fair
**Rating:** 4
**Confidence:** 4

**Summary:**

This paper is about the exploration of using pseudolabels for VLMs on different downstream tasks. Specifically, the authors experimented CoOp, VPT and UPT with pseudolabels generated from CLIP on semi-supervised learning, transductive zero-shot learning and unsupervised learning. They proposed three training strategies, namely FPL, IFPL and GRIP, where the difference is mainly at whether the pseudolabels are static or dynamic. Experiments on 6 datasets show significant improvement.

**Strengths:**

- Writing is good. The work is presented in a logical way and is easy to follow.
- Performance improvement is impressing.
- Experiments are thorough.

**Weaknesses:**

- Novelty is limited. The use of CLIP's pseudolabels are not new and the proposed training strategies are also widely used in self-training methods.
- I agree with the authors' statement that "If CLIP performs poorly on a task, we may struggle to obtain a reliable set of pseudolabels to begin with, potentially diminishing CLIP’s performance." Therefore, it is unintuitive why GRIP achieves such a big improvement even on datasets like MNIST where CLIP fails. One explanation I can think of is that the quality of the pseudolabels are drastically increasing as the training proceeds. However, results from Fig 3 seems to show the opposite. I am mainly concerned about this and believe further analysis is needed to explain what is the exact reason for the success of the proposed method.
- I'm also concerned with the computational cost since a total of 10 iterations are used. Authors should include this when comparing with other methods.

**Questions:**

- The authors stated that the proposed IFPL and GRIP are "novel and proposed here for the first time for prompt tuning". However, aren't these two methods similar in spirit to self-training? What are the differences?
-  According to Fig 3, the accuracy of pseudolabels are only marginally increasing for IFPL, and decreasing for GRIP as the training proceeds. Is it necessary then to train over 10 iterations? What are the detailed improvements for each iteration?
- Why is it that CLIP's pseudolabel accuracy decreases in Fig 3? Shouldn't zero-shot CLIP remain an approximately constant performance irrespective of the number of iteration and the number of unlabeled data?
- When using a larger image encoder, why does the performance decrease? If a larger backbone is used, wouldn't the quality of pseudolabels be higher? This further deepens my concerns on the proposed method.

**Limitations:**

Yes, the authors have addressed the limitations.

---

> ### Author Rebuttal · Authors · 2023-08-06
>
> Thank you for your thoughtful feedback. Below we address all your questions.
>
> > Novelty is limited. The use of CLIP's pseudolabels are not new and the proposed training strategies are also widely used in self-training methods.
>
> We respectfully disagree that novelty is limited. Our study centers on conducting a comprehensive exploration of a wide design space that includes prompt modalities, learning paradigms, and training strategies. As shown in Figure 1, this design space has been an underexplored topic in the literature. If there are other papers that have considered these choices in the context of vision-language models like CLIP, we would be grateful for pointers. As also discussed above in the general comment, this exploration leads to novel and useful findings about prompt tuning across a wide range of settings like UL and TRZSL including improved performance and the Robin Hood effect.
>
> > I agree with the authors' statement that "If CLIP performs poorly on a task, we may struggle to obtain a reliable set of pseudolabels to begin with, potentially diminishing CLIP’s performance." Therefore, it is unintuitive [...] I am mainly concerned about this and believe further analysis is needed to explain what is the exact reason for the success of the proposed method.
>
> The concerns raised by the reviewer about MNIST can be clarified in Figure 6 of the Appendix. As the plots display, for SSL and UL the accuracy of GRIP pseudolabels is pretty high and maintained throughout the iterations. Moreover, its value is close to the accuracy of MNIST reported in Table 1.  As for TRZSL, the accuracy is close to the one of CLIP and in absolute value reflects the overall accuracy reported in Table 1. We explain the fact that GRIP accuracy is higher than CLIP, even with comparable pseudolabels accuracy, with the trade-off between quality and quantity that we extensively cover in Section 4.1, page 7, and the Appendix.
>
> > I'm also concerned with the computational cost since a total of 10 iterations are used. Authors should include this when comparing with other methods.
>
> And
>
> > According to Fig 3, the accuracy of pseudolabels are only marginally increasing for IFPL, and decreasing for GRIP as the training proceeds. Is it necessary then to train over 10 iterations? What are the detailed improvements for each iteration?
>
> Repeating the training process multiple times brings impressive improvements at the cost of a non-negligible increase of computation time. While we parallelized the pseudolabeling procedure to cut some of the cost, reducing those for iterative training presents more significant challenges. We decided to mainly focus on the analysis of qualitative and quantitative effects of pseudolabels in prompt tuning, as overviewed in Figure 1.
>
> Future research should address budget constraints and, as suggested by the reviewer, investigate optimal stopping criteria for the iterative process, considering the possibility of reaching a plateau or decreased pseudolabel quality after a certain point, to maximize efficiency while maintaining performance.
>
> Due to the importance of the topic, we will add a discussion of computational time and cost in the limitations section of the paper.
>
> > The authors stated that the proposed IFPL and GRIP are "novel and proposed here for the first time for prompt tuning". However, aren't these two methods similar in spirit to self-training? What are the differences?
>
> IFPL and GRIP are novel in the context of prompt tuning and CLIP-based pseudolabels since they had not been previously investigated. However, it is fair to say that they are similar in spirit to the common self-training approach. The key distinction is the pseudolabel assignment method: using a top-K strategy instead of a confidence threshold. Despite the subtlety of this difference, it significantly impacts the final prediction’s quality. Our experiments demonstrate that prompt tuning with IFPL and GRIP achieves a more equitable distribution of class accuracies.
>
> If considered misleading, we can replace “novel” with “unexplored”.
>
> > Why is it that CLIP's pseudolabel accuracy decreases in Fig 3? Shouldn't zero-shot CLIP remain an approximately constant performance irrespective of the number of iteration and the number of unlabeled data?
>
> When we increase the amount of pseudolabeled data, the accuracy of CLIP is not necessarily supposed to remain constant. As we increase K, we are effectively selecting pseudolabels with lower similarities to the classes, resulting in a reduction in their accuracy, as shown in the plot. This observation aligns with previous findings in [15].
>
>
> > When using a larger image encoder, why does the performance decrease? If a larger backbone is used, wouldn't the quality of pseudolabels be higher? This further deepens my concerns on the proposed method.
>
> We would like to clarify that when using a larger encoder, GRIP still substantially improves the performance across all learning paradigms.  The smaller relative improvements with respect to smaller visual encoders align with our expectations. Larger encoders possess a stronger base knowledge, making it relatively more challenging to attain further improvements on top of it.
>
> In Table 10 (additional PDF for rebuttal, which we will add to an appendix), we report the accuracy of CLIP with different backbones. To clarify, the performance of the larger backbone is higher, indicating higher quality pseudolabels as the reviewer says. Table 3 is included to show that techniques like GRIP generalize across backbones, but it is unsurprising that the gains are a bit smaller with a larger backbone because it is a stronger starting point.

---

> > ### Comment · Reviewer_qyTk · 2023-08-21
> > **Response to rebuttal**
> >
> > Dear authors,
> >
> > Thank you for the feedback! Most of my concerns were addressed (e.g., performance on MNIST and with a stronger backbone). I've also carefully read other reviewers' comments and the authors' responses. However, since pseudolabeling of CLIP is already widely explored in the unsupervised setting, I still feel that extending such methodology to SSL and TZSL is a marginal step. The experiments are good, but I would prefer seeing something new in the design. Therefore I would remain my score.

---

> > > ### Author Response · Authors · 2023-08-21
> > >
> > > Thank you for your response, we are glad your main concerns have been solved.
> > >
> > > For the sake of the final discussion among reviewers, we provide some clarifications about the goals of our work.
> > >
> > > > “since pseudolabeling of CLIP is already widely explored in the unsupervised setting”
> > >
> > > In our design space, for unsupervised learning we define 6 paths to explore, among which only 1 was previously investigated in the literature [15]. If there are other papers that have considered these choices in the context of vision-language models like CLIP, we would be grateful for pointers.
> > >
> > > > “I still feel that extending such methodology to SSL and TZSL is a marginal step”
> > >
> > > We respectfully disagree with the reviewer. We do not claim to propose or extend methodologies to work in the SSL and TRZSL settings, rather we extensively explore how pseudolabeling adapts to these settings in the context of prompt tuning with CLIP.
> > >
> > > The exploration of our underexplored design space is based on the observations that (1) VLMs' zero-shot capabilities extend the usability of pseudolabels to any limited-label data scenario, and (2) paradigms such as semi-supervised, transductive zero-shot, and unsupervised learning can all be seen as optimizing the same loss function, by using zero-shot pseudolabels as a source of supervision.
> > >
> > > We consider these two observations, and the experimental findings that generalize across these settings, as important takeaways from this paper.
> > >
> > > > “The experiments are good, but I would prefer seeing something new in the design”
> > >
> > > The goal of this paper is to study how to use pseudolabels with CLIP in a variety of unexplored learning settings. We do not propose new methodologies to use pseudolabels with CLIP.
> > >
> > > We explored 27 combinations of prompt modalities, learning settings, and training strategies, among which only 1 was already explored. We consider filling this gap in literature a significant contribution that shows useful takeaways to the community.
> > >
> > > In this context, our experiments not only demonstrate the effectiveness of using pseudolabels iteratively for prompt tuning CLIP in limited labeled scenarios, but also show that prompt tuning with pseudolabels can mitigate the biases of  the original model.
> > >
> > > Reviewers generally appreciated our contributions:
> > > * Reviewer sr7c says: “ The authors provide compelling evidence that underlines the power of a repetitive prompt-training approach, which leverages CLIP-based pseudo labels. Regardless of the learning model (SSL, TZSL, UL) or the type of prompt (text, visual), this strategy significantly enhances the image classification capabilities of CLIP across multiple settings. [..] effectively addressing the inherent bias of CLIP pseudo labels.”
> > > * Reviewer kGE6 says: “ agreeing with the position of Reviewer 1u3c, that it is a necessary milestone in the research community for under-explored CLIP pseudo-labeling which can stimulate other works and deeper understanding of pseudo-labeling algorithm aspects.”
> > >
> > > * Reviewer qyTk: “ Performance improvement is impressing. Experiments are thorough”

---

### Official Review · Reviewer_sr7c · 2023-07-07

**Soundness:** 2 fair
**Presentation:** 2 fair
**Contribution:** 2 fair
**Rating:** 6
**Confidence:** 4

**Summary:**

The authors provide compelling evidence that underlines the power of a repetitive prompt-training approach, which leverages CLIP-based pseudo labels. Regardless of the learning model (SSL, TZSL, UL) or the type of prompt (text, visual), this strategy significantly enhances the image classification capabilities of CLIP across multiple settings. Moreover, by using the Top-K pseudo-labeling approach, they ensure a balanced distribution of pseudo-labeled training samples for each class, thereby effectively addressing the inherent bias of CLIP pseudo labels.

**Strengths:**

- Various learning paradigms, including semi-supervised, transductive zero-shot, and unsupervised learning, can all be viewed as unique instances of a single objective function when pseudolabels are used as a form of guidance.
-  As evidenced by Table 1and Table 2, the experimental results demonstrate a significant boost in performance, indicating that the proposed pseudo-labeling strategy holds considerable merit.

**Weaknesses:**

The overall structure or layout of the manuscript needs to be further refined.
- Table 5 ("There is a trade-off between quality and quantity of pseudolabels") is mentioned in L290, but Table 5 does not exist in the main manuscript.
- Table 2 was not mentioned at all in the body text.
- Since Fig 3 and Fig 5 are too small to be seen properly, efforts are needed to improve readability.
- The formulas of L157-L158 should be specified as an equation in the form of (1), but it was not.

Methodologically, the proposed approach is too naive.
- Setting the trade-off parameters in unified objective function (L157-L158), e.g., gamma, and lambda, is too heuristic, and there is lack of the analysis.
- Selecting the Top-K most confident samples by class is too naive. It would be nice to select a confident sample by improving class diversity through the application of techniques such as information maximization loss [1]. And it would be good to present comparison results with other attempts for class diversity.

[1] Liang, Jian, Dapeng Hu, and Jiashi Feng. "Do we really need to access the source data? source hypothesis transfer for unsupervised domain adaptation." ICML 2020

**Questions:**

- In Table 1 and Table 2, there's a noticeable drop in some experimental results under the UL setting. I'm curious as to the cause of this decrease.
- Within Table 2, the outcomes for the TRZSL learning setting in RESICS45 and DTD, under the "Multimodal prompts" category, show a decline. I'm interested in understanding why this is so.
- In the experimental results presented in Table 3, where "GRIP benefits adaptation even for larger image encoders," there seems to be a reduced rate of improvement when using a larger image encoder. I wonder the reason or analysis behind this observation.

**Limitations:**

I already pointed out in the weakness part.

---

> ### Author Rebuttal · Authors · 2023-08-06
>
> Thank you for your thoughtful feedback. Below we address all your questions.
>
> About the layout and typos we will address all the points in the final version of the paper.
>
> > Setting the trade-off parameters in unified objective function (L157-L158), e.g., gamma, and lambda, is too heuristic, and there is lack of the analysis.
>
> Indeed the trade-off parameters we defined are heuristic, but they have proven to be effective across all 162 combinations of tasks, prompt modalities, learning paradigms, and training strategies that we extensively investigated.
>
> Although having a theoretical analysis is always desirable, it is worth emphasizing that empirically setting hyperparameters to balance the terms of a loss function is the common practice in machine learning. Often, hyperparameter optimization can be resource-intensive, whereas in our approach, equally weighting labeled and pseudolabeled data yields remarkable improvements and obviates the need for an exhaustive hyperparameter search.
>
> > Selecting the Top-K most confident samples by class is too naive. It would be nice to [..] other attempts for class diversity.
>
> We respectfully disagree in considering simplicity a weakness, especially after observing the quantitative (impressive performance improvement) and qualitative advantages (Robin Hood effect) of using the top-K approach for pseudolabeling. Since we set our goal to explore how to use CLIP’s pseudolabels in a variety of low-resource settings for prompt tuning, we deliberately decided to focus on the design space overviewed in Figure 1. Using more complex procedures for other aspects of the experiments risks confounding the results.
>
> > In Table 1 and Table 2, there's a noticeable drop in some experimental results under the UL setting. I'm curious as to the cause of this decrease.
>
> The drops in performance under the unsupervised learning (UL) setting typically correspond to a poor initial assignment of pseudolabels, perpetuated through the iterative process. Indeed, in UL pseudolabeled data is the only source of supervision. However, this phenomenon can also be attributed to the prompt modality, as we can observe improvements under the same setting where a prompt of different modality is trained.
>
> > Within Table 2, the outcomes for the TRZSL learning setting in RESICS45 and DTD, under the "Multimodal prompts" category, show a decline. I'm interested in understanding why this is so.
>
> We speculate that during the first training iteration, the learned prompts produce low-quality pseudolabels, leading to suboptimal learning in subsequent iterations. In contrast, GRIP - which increases the set of pseudolabels over time - significantly outperforms the baseline in these cases.
>
> Understanding why this happens for multimodal prompts but not for textual or vision prompts is challenging. In the literature, the behavior of different prompt modalities is typically associated with the task’s characteristics [44]. However, the prompt tuning literature lacks a definitive scientific consensus on the effectiveness of one modality over another.
>
> > In the experimental results presented in Table 3, where "GRIP benefits adaptation even for larger image encoders," there seems to be a reduced rate of improvement when using a larger image encoder. I wonder the reason or analysis behind this observation.
>
> The smaller relative improvements with respect to smaller visual encoders align with our expectations. Larger encoders possess a stronger base knowledge, making it relatively more challenging to attain further improvements on top of it.
>
> In Table 10 (additional PDF for rebuttal, which we will add to an appendix), we report the accuracy of CLIP with different backbones. To clarify, the performance of the larger backbone is higher, indicating higher quality pseudolabels. Table 3 is included to show that techniques like GRIP generalize across backbones, but it is unsurprising that the gains are a bit smaller with a larger backbone because it is a stronger starting point.

---

> > ### Comment · Reviewer_sr7c · 2023-08-21
> > **Response to Rebuttal**
> >
> > Thanks for addressing my feedback; I'm inclined to increase my score from 4 to 6 in favor of the paper's acceptance.

---

> > > ### Author Response · Authors · 2023-08-21
> > >
> > > Thank you for your feedback and the support!
> > >
> > > We will include all the necessary clarification in the final version of the paper!

---

### Official Review · Reviewer_kGE6 · 2023-07-07

**Soundness:** 3 good
**Presentation:** 3 good
**Contribution:** 3 good
**Rating:** 7
**Confidence:** 4

**Summary:**

In this paper authors extend empirical study of self-training for the case when pseudo-labels are generated by models (CLIP) in zero-shot regime (models are trained on unlabeled data with respect to a downstream task, but can be used for zero-shot prediction for the downstream task). Authors investigate self-training in case of visual-language models (CLIP) with tuning visual/text/visual-text prompts with pseudo-labels generated in the zero-shot regime by CLIP model. They formulate also different training regimes (unsupervised, semi-supervised, transductive zero-shot) in terms of simpler supervised training where part of data (or all) just has pseudo-labels. Also it is investigated how we could bootstrap further performance in self-training doing rounds of training while updating the pseudo-labels. In the end authors show that self-training is effective and improve across different settings and datasets. Interesting results are obtained in terms of debiasing original CLIP model when self-training is done afterwards.

**Strengths:**

- extensive empirical study of self-training for the CLIP-based models and various configurations and settings
- demonstrating that CLIP-style models can be used as seed model for self-training in variety of settings bringing significant boost for prompt finetuning
- empirical demonstration how CLIP de-biasing can be done via self-training

**Weaknesses:**

- absence of any ablations on balancing between labeled / unlabeled data
- absence of ablation on the number of samples selected for every class in self-training (why K=16? is it important? why balanced across classes?), also it is not clear what is the effect of balanced selection here especially on the de-biasing the CLIP (if de-biasing happens only because of this balancing then it is very obvious/expected result in my opinion)
- GRIP exact explanation / how it works should be extended in the text (many ambiguities in the current description)
- Table 1 and other Tables: I don't get why for CLIP TRZSL is similar to SSL and UL while for self-training in any combination TRZSL becomes way better than SSL and UL?
- Missed discussion on the results that text prompt alone gives superior results compared to text+visual prompt or visual-alone prompt

**Summary** There is no novelty or any impactful design of the self-training algorithm itself, though extensive empirical study is done for zero-shot learners (CLIP) used to pseudo-label data. The particular choice of hyper-parameters (like balancing pseudo-labels per class) demonstrate significant improvement for the prompt fine-tuning task as well as de-biasing original model (latter I found particularly important and interesting).

**Questions:**

- lines 50-53: I would not agree that VLM are trained on unlabeled data. We do mining of audio-visual pairs, so in that sense it is really unlabeled. On the other hand, yes, downstream task is different and we can use zero-shot inference to generate pseudo-label. However one could consider that task of pairing text and audio is more general than pairing class name and image. I would be here more concrete in phrasing.
- There are a lot of repetitions of the same sentences / phrases along the text, e.g. lines 60-69 (other - lines 134-144) appear couple of times. I would reorganize the text to refer to the previous parts rather than repeat almost same ways in introduction / results / sections.
- typo line 96: "such as such as"
- lines 179 and 187 - selection of $gamma$ and $\alpha$ seems to be equivalent providing 1:1 balancing between labeled / unlabeled data. I don't get why then expression are made different if in the end mathematically optimization is the same.
- IFPL: what happens if for some class we don't have K samples? say no any examples are predicted as class 0 for MNIST, what happens then?
- How about IFPL variant but when we increase K over time, still taking top only pseudo-labels? This is I assume is different from the GRIP.
- It is not clear if in GRIP classes for pseudo-labels are still balanced, and also how do we select which part of unlabeled set to take? Do we take every iteration top pseudo-labels increasing the number of them or we just increase randomly the data size and take whatever pseudo-labels will be there? Do we still re-initialize prompt every iteration here? Do we reuse or regenerate pseudo-labels for the part of data used in previous iteration here?
- Why we don't train continuously prompt when we re-generate pseudo-labels? Why do we not fine-tune the whole model also but only prompting (if for efficiency purpose - ok, but interesting to see if we need to finetune the whole model)?
- lines 308-311 - this is actually known fact. E.g. for fixmatch/remixmatch we do different augmentations, and they are the key to make self-training to work, so particular type of noise in data/labels make self-training to work.
- Figure 3 (and similar in Appendix) is not clear. I got what is x-axis only after reading it several times. Either better notation or better caption is needed. Maybe "refers to the top x-axis (number of iterations) while .. to the bottom a-axis (amount of unlabeled data)".
- I still don't get why GRIP over iterations can become worse than IFPL. Also maybe it is worth to have ablation where we do IFPL but for every iteration K is increased so that we increase amount of unlabeled data involved though we do this over the most confident every time still.
- Robin Hood effect: is it because we do balancing of classes in self-training? I guess here this plays a huge role, assuming that some classes are unrepresented in CLIP pretraining + we know that in self-training different classes have different pace of learning, so that balancing can resolve issues on under representative classes or hard ones.
- line 345 - what is "conventional pseudo-labeling"?
- Table 5: typo - UPL -> UPT
- lines 664-665 - is it because of the CLIP bias itself and the way classes are balanced in fine-tuning?
- lines 692-693 - maybe not surprising as we balanced examples per class?

**Limitations:**

Limitations are discussed after conclusion sections. Formulation sounds reasonable to me.

**Update: after rebuttal and discussion with authors the score is update from 6-weak accept to 7-accept and contribution from 2 to 3.**

---

> ### Author Rebuttal · Authors · 2023-08-06
>
> Thank you for your thoughtful feedback. Below we address all your questions that space allows. We do not seem to be able to reply to our own rebuttal to answer more. Please reply if you would like us to answer the remaining questions in our own reply.
>
> > Absence of any ablations on balancing between labeled / unlabeled data
>
> We started experimenting without balancing labeled and unlabeled data. This yielded unsatisfactory results  since the training excessively focused on either of the two sets of data. So, we tried balancing the two. This strategy turned out to robustly work for all the 162 combinations of datasets, prompt modalities, learning settings, and training strategies we explored. We observed impressive quantitative and qualitative performance improvements, while eliminating the need for a hyperparameter search to optimize results. We decided not to confound the conclusions  with balance optimization due to our focus on the design space overviewed in Figure 1.
>
> > Absence of ablation on the number of samples selected for every class in self-training (why K=16? is it important? why balanced across classes?)
>
> We set K=16 since it is indicated as the optimal K in the previous research on pseudolabeling with CLIP [15]. In general, K is an hyperparameter that that may require optimization in practical cases. We decided to be consistent with the literature and applied this fixed value of K in order to focus on the design space overviewed in Figure 1.
>
> The balance across classes also derives from [15]. About that, we note that this is an easy and effective way to avoid pseudolabels distribution skewed toward certain classes.
>
> > Table 1 and other Tables: I don't get why for CLIP TRZSL is similar to SSL and UL while for self-training in any combination TRZSL becomes way better than SSL and UL?
>
> CLIP's performance similarity across TRZSL, SSL, and UL is due to evaluation on the same test set. While SSL and UL get the same scores, TRZSL accuracy differs since it is the harmonic mean between seen and unseen class accuracies. This helps recalibrate the overall score, particularly when the model performs poorly on one set of classes.
>
> Observing significantly larger scores for TRZSL compared to SSL and UL is expected.  This difference arises because in TRZSL, a portion of the target classes is provided with labeled data, which contributes to a larger ground truth being available to the model compared to SSL and UL settings.
>
> > Missed discussion on the results that text prompt alone gives superior results compared to text+visual/visual prompt
>
> Using pseudolabels dynamically is beneficial for each modality. However, determining the clear superiority of one prompt modality over the other is challenging, as it depends on the specific tasks. For example, visual prompts work better for EuroSAT, while textual prompts excel in Flowers102. Despite intuitive explanations (Section 3.1), the scientific consensus remains elusive [44]. Hence, we prefer to emphasize that the dynamic use of pseudolabels consistently improves performance for each prompt modality, without declaring one modality as definitively better than the other.
>
> > The selection of lambda and gamma seems to be equivalent providing 1:1 balancing between labeled / unlabeled data. I don't get why then expression are made different if in the end mathematically optimization is the same.
>
> Yes, they are equivalent. We will simplify the presentation and just leave the one weighting the unlabeled data.
>
> > IFPL: what happens if for some class we don't have K samples? say no any examples are predicted as class 0 for MNIST, what happens then?
>
> The top-K pseudolabeling strategy we adopt, originally studied in [15], consists of (1) computing the similarity scores of each datapoints with classes’ textual prompts, and  (2) select for each class the K datapoints with the highest similarity score to the class. In this way, we always get K pseudolabels per class. Of course, it can happen that if K is too large and the unlabeled dataset is smaller than K*|number of classes|. In this case, we suggest reducing K.
>
> > How about IFPL variant but when we increase K over time, still taking top only pseudo-labels? This is I assume is different from the GRIP.
>
> Based on our understanding, it appears that the variant of IFPL proposed by the reviewer is the same as GRIP. In IFPL, top-K pseudolabels are recomputed at each iteration while keeping K fixed. In contrast, in GRIP, K increases with each iteration and we assign pseudolabels still following the top-K schema.
>
> > It is not clear if in GRIP classes for pseudo-labels are still balanced, and also how do we select which part of unlabeled set to take? [...] Do we reuse or regenerate pseudo-labels for the part of data used in previous iteration here?
>
> GRIP maintains class balance by selecting the top-K samples at each iteration, with K increasing progressively. Similar to IFPL, both prompts and pseudolabels are reinitialized with every iteration, in order to avoid accumulating errors from earlier iterations. In other words, learning progresses from pseudolabels to new prompts to new pseudolabels, and so on.
>
> > Why we don't train continuously prompt when we re-generate pseudo-labels? Why do we not fine-tune the whole model [...]?
>
> We reinitialized prompts at each iteration to avoid error accumulation from previous steps. The focus of this work is on parameter-efficient adaptation of CLIP, thus we did not try out the use of pseudolabels to fine-tune the entire model. Additionally, it is worth noting that the limited amount of initial training data might not be sufficient to meaningfully fine-tune such a large model.
>
> We will clarify this point in the revised version of the paper.
>
> > I still don't get why GRIP over iterations can become worse than IFPL. [...].
>
> As reported in Table 2, GRIP either outperforms or performs comparatively with IFPL. The suggestion of the reviewer is correct and it corresponds to GRIP.

---

> > ### Comment · Reviewer_kGE6 · 2023-08-15
> > **Reviewer's response to rebuttal**
> >
> > Dear authors,
> >
> > Thanks a lot for detailed clarifications. I have carefully read all reviews and your responses. I strongly suggest you to incorporate all main points of the discussion into the final revision (e.g. the choice of balancing and top-k of 16). I still have some questions and comments to clarify:
> >
> > >  Absence of any ablations on balancing between labeled / unlabeled data
> >
> > Ok, I buy your arguments here. But I think you should include this into text to provide for the future as reference of observations.
> >
> > > Absence of ablation on the number of samples selected for every class in self-training (why K=16? is it important? why balanced across classes?)
> >
> > As you have reference to prior work - nice to see it still works confirming robustness of prior work choice. Add this to the text as it reasons on the choice you made.
> >
> > > Table 1 and other Tables: I don't get why for CLIP TRZSL is similar to SSL and UL while for self-training in any combination TRZSL becomes way better than SSL and UL?
> >
> > I think this clarification should be included in the final revision to make readability clear.
> >
> > >  (2) select for each class the K datapoints with the highest similarity score to the class.
> >
> > Could then we end up having some samples presented in data with two different assigned pseudo-labeled classes?
> >
> > > Based on our understanding, it appears that the variant of IFPL proposed by the reviewer is the same as GRIP. In IFPL, top-K pseudolabels are recomputed at each iteration while keeping K fixed. In contrast, in GRIP, K increases with each iteration and we assign pseudolabels still following the top-K schema.
> > > GRIP maintains class balance by selecting the top-K samples at each iteration, with K increasing progressively. Similar to IFPL, both prompts and pseudolabels are reinitialized with every iteration, in order to avoid accumulating errors from earlier iterations. In other words, learning progresses from pseudolabels to new prompts to new pseudolabels, and so on.
> >
> > From description in line 223 I do not agree with your statement. "we use i/I -th of the unlabeled data" - there is nothing about top-K where K is growing. You need to be more precise in the description of GRIP in the text for the revision. But thanks for the explanation, now it makes sense to me :) (And general response answers entirely my concern here).
> >
> > Any comment on
> > > Robin Hood effect: is it because we do balancing of classes in self-training? I guess here this plays a huge role, assuming that some classes are unrepresented in CLIP pretraining + we know that in self-training different classes have different pace of learning, so that balancing can resolve issues on under representative classes or hard ones.
> >
> > ?
> >
> > > As reported in Table 2, GRIP either outperforms or performs comparatively with IFPL. The suggestion of the reviewer is correct and it corresponds to GRIP.
> >
> > Here I look e.g. at Fig 6 in Appendix where accuracy drops over iterations for GRIP but not for IFPL. I found this opposite to the expected behaviour where GRIP should improve things over iterations as we become more confident (maybe this shows overfiting). Do you report last iteration performance for GRIP or the best in Table 2?
> >
> > Thanks,
> > Reviewer kGE6.

---

> > > ### Author Response · Authors · 2023-08-16
> > >
> > > Thank you again for all the suggestions. They are all valuable and clarifications will certainly be included in the final version of the paper.
> > >
> > > Below, we answer further questions and post the answers we wrote and left out because of space constraints.
> > >
> > > > (2) select for each class the K datapoints with the highest similarity score to the class. [..] Could then we end up having some samples presented in data with two different assigned pseudo-labeled classes?
> > >
> > > Yes, this can happen. However, we checked the pseudolabels at each iteration and it was rarely the case.  This is a characteristic of the pseudolabeling strategy proposed in [15]. We believe this can be an object of study for future work motivated by the effectiveness of self-training.
> > >
> > > > Robin Hood effect: is it because we do balancing of classes in self-training? I guess here this plays a huge role, assuming that some classes are unrepresented in CLIP pretraining + we know that in self-training different classes have different pace of learning, so that balancing can resolve issues on under representative classes or hard ones.
> > >
> > > In our paper, we conducted a comprehensive investigation into the causes of the Robin Hood effect, which results in a more balanced distribution of class accuracies (Section 4.2). Our study revealed that the combination of the top-K pseudolabeling strategy and prompt tuning plays a crucial role in achieving this effect. We hypothesize that the parameter-efficient nature of prompt tuning also helps avoid overfitting to the easier classes.
> > >
> > > Surprisingly, this straightforward and intuitive approach, aimed at balancing classes, has not been explored in previous works that address imbalanced accuracy distributions, as evidenced in references [49,8]. While these studies emphasized the importance of examining class accuracies when employing pseudolabels in semi-supervised learning, we extend this analysis to the application of prompt tuning and CLIP-pseudolabeling. By doing so, we present a novel perspective on this phenomenon and its practical implications.
> > >
> > > > Here I look e.g. at Fig 6 in Appendix where accuracy drops over iterations for GRIP but not for IFPL. I found this opposite to the expected behaviour where GRIP should improve things over iterations as we become more confident (maybe this shows overfiting). Do you report last iteration performance for GRIP or the best in Table 2?
> > >
> > > We report the performance of the last iteration. The behavior of GRIP’s pseudolabels accuracy surprised us too. That is why we emphasize it by discussing the trade-off between the quantity and quality of pseudolabels (Sect 4.1, page 7). Although the accuracy of GRIP decreases we can still observe that it is higher than the accuracy of CLIP on the same amount of data. Moreover, the accuracy of pseudolabels at the last iteration is close to the overall accuracy of GRIP in Table 1 and 2.
> > > We speculate that one of the causes of the deterioration of psuedolabels accuracy could be overfitting, but we do not have evidence to entirely support this claim and leave the question open for further explorations.
> > >
> > > > lines 308-311 - this is actually known fact. E.g. for fixmatch/remixmatch we do different augmentations, and they are the key to make self-training to work, so particular type of noise in data/labels make self-training to work.
> > >
> > > In lines 308-311: we say “This suggests that numerous, slightly noisier pseudolabels can yield better results, highlighting a trade-off and offering insights for future approaches.”
> > >
> > > We clarify that the word choice of “noisier” is misleading and led the reviewer to think we are referring to augmentations. Instead, with “noisier” we refer to incorrectly assigned pseudolabels. Specifically, in that sentence we explain the trade-off between quality and quantity of pseudolabels.
> > >
> > > To avoid confusion we will replace the word “noisier” with “incorrect”.
> > >
> > > > lines 692-693 - maybe not surprising as we balanced examples per class?
> > >
> > > The class balance is imposed on both methods we compare. However, the class accuracy distribution of CLIP via linear probing shows a 30 times larger reduction in the accuracy of rich classes. Lines 692-693 comment on the class accuracy distribution obtained running GRIP with linear probing.  While the class balance plays a central role to determine the Robin Hood effect, in this context we were commenting that with linear probing the reduction of accuracy of rich classes is on average 30 times larger than the reduction observed using prompts.
> > >
> > > > lines 664-665 - is it because of the CLIP bias itself and the way classes are balanced in fine-tuning?
> > >
> > > In the lines highlighted, we observe that the accuracy of GRIP on seen classes is worse than CoOp. During training, for large lambda, the loss component of unlabeled data (unseen classes) is the first to decrease, while the loss on the seen classes reduces later. Thus, we hyphotize that extra training steps might be needed to complete the learning on the labeled data.

---

> > > > ### Comment · Reviewer_kGE6 · 2023-08-19
> > > > **Final comments and raising score**
> > > >
> > > > Dear authors,
> > > >
> > > > Sorry for the delay in response. Thanks for providing more answers to my questions. Just one last comment from my side:
> > > >
> > > > > We report the performance of the last iteration. The behavior of GRIP’s pseudolabels accuracy surprised us too. That is why we emphasize it by discussing the trade-off between the quantity and quality of pseudolabels (Sect 4.1, page 7). Although the accuracy of GRIP decreases we can still observe that it is higher than the accuracy of CLIP on the same amount of data. Moreover, the accuracy of pseudolabels at the last iteration is close to the overall accuracy of GRIP in Table 1 and 2. We speculate that one of the causes of the deterioration of psuedolabels accuracy could be overfitting, but we do not have evidence to entirely support this claim and leave the question open for further explorations.
> > > >
> > > > Yep, I could look into overfitting as first thing by checking how many times the same sample participated in training, as then it could be that we just overfit to the top-1 sample during the whole training if they appear in every GRIP iteration.
> > > >
> > > > Summary: after discussion most of my concerns were resolved, and the ones which are still there (e.g. some extra ablations) I don't think are so important regarding paper acceptance. I recommend the paper for acceptance, agreeing with the position of Reviewer 1u3c, that it is a necessary milestone in the research community for under-explored CLIP pseudo-labeling which can stimulate other works and deeper understanding of pseudo-labeling algorithm aspects. About the computational budget: this argument as the weakness is not strong for me as there are a bunch of methods developed in many areas, like vision, MT and speech to have computationally efficient pseudo-labeling (when we train one model with regenerating pseudo-labels during training) -- so here I would not be surprised that they are applicable with small modifications and could resolve the computational budget issue.
> > > >
> > > > I would like to raise my score from 6 to 7 to support the paper acceptance, though discussion should be incorporated in the final revision.
> > > >
> > > > Thanks again for productive discussion.

---

> > > > > ### Author Response · Authors · 2023-08-19
> > > > >
> > > > > Thank you so much for all your valuable input and for supporting our paper! Thank you as well for the productive discussion. We will be sure to incorporate the great feedback into the final version.

---

### Official Review · Reviewer_1u3c · 2023-07-09

**Soundness:** 3 good
**Presentation:** 3 good
**Contribution:** 2 fair
**Rating:** 6
**Confidence:** 3

**Summary:**

The paper explores the use of CLIP for pseudo-labeling for various tasks such as SSL and on various datasets. Overall - while not being very surprising - the results can clearly outperform prior work (that is not using CLIP it has to be said) - and thus showing the potential of CLIP for such tasks.

**Strengths:**

Clearly, the authors show what they set out to do: CLIP pseudo-labeling can improve via pseudo-labeling on a variety of tasks and datasets. That is indeed nice to see.

The positive experiments are good - doing this for a variety of tasks and on a variety of datasets.


**Weaknesses:**

While the improvements are good - they are not surprising either. CLIP is trained on a wide variety of data and thus it is clear that pseudo-labeling using CLIP should help in many cases

In my view the main weakness of the paper is that it is not showing the limits of CLIP pseudo-labeling - while most of us do not know which exact data is used to train CLIP - the datasets used seem still reasonably close.

**Questions:**

Overall the paper has a clear point and it does a good job making that point reasonably clear. As said above, after reading the paper I am not surprised that this can work, but I am left without an answer what the limits are of CLIP pseudo-labeling,



**Limitations:**

limitation discussion ok for me - except the point of the limits of CLIP pseudolabeliing as mentioned above

---

> ### Author Rebuttal · Authors · 2023-08-06
>
> Thank you for your thoughtful feedback. Below we address all your questions.
>
> > The results can clearly outperform prior work (that is not using CLIP it has to be said) - and thus showing the potential of CLIP for such tasks.
>
> We would like to clarify that  the baselines and comparisons described in LL237-242 are using CLIP. The wide range in performances show the benefits of pseudolabeling and the importance of the choices within the design space we explore.
>
> > While the improvements are good - they are not surprising either. CLIP is trained on a wide variety of data and thus it is clear that pseudo-labeling using CLIP should help in many cases
>
> Because of the remarkable zero-shot ability of CLIP, we agree about the possibility of getting sufficiently good pseudolabels. However, we respectfully disagree about the clear benefit we can derive from them. Indeed, in our paper we show that the naive and static usage of top-K pseudolabels per-class, as proposed in [15], does not fully unlock their potential. On the contrary, dynamic training strategies show good improvements.
>
> > In my view the main weakness of the paper is that it is not showing the limits of CLIP pseudo-labeling while most of us do not know which exact data is used to train CLIP - the datasets used seem still reasonably close.
>
> We agree that exploring the limitations of a method is also important. In this work, we were surprised to find that pseudolabeling can be effective for improving CLIP on such a wide range of tasks. We ran our experiments on domain-specific tasks where CLIP exhibited poor performance, primarily attributed to domain shift issues, as highlighted by CLIP's authors [31, Section 3.1.5, and Figure 5]. It was surprising to see that pseudolabels can bring improvements even in cases where CLIP has low baseline performance, e.g., EuroSAT, DTD, and FGVCAircraft.
>
> We know that pseudolabeling will fail if CLIP’s initial performance is not much better than random guessing or adversarially biased. We have discussed this in the paper’s limitations section. But given the range of datasets and design choices already considered, we had to leave searching for additional datasets on which it fails for future work. Perhaps datasets with very different sensors, like medical imaging, would be even more challenging.

---

> > ### Comment · Reviewer_1u3c · 2023-08-18
> >
> > After reading all reviews and the authors' rebuttal I personally still lean towards acceptance of the paper. While I agree that one could argue that methodologically there is not much novelty, I still consider it worth reporting given the reported improvements and the fact tat CLIP pseudolabeling is still rather under-explored.
> >
> > I also fully agree with on of the other reviewers that the authors should included the discussions and clarifications mentioned in the reviews and rebuttal to strengthen the paper overall.

---

> > > ### Author Response · Authors · 2023-08-18
> > >
> > > We are glad the reviewer appreciated our work and emphasized the paper's contribution as extending beyond methodological novelty.
> > >
> > > Thank you again for your review. We will make sure to add valuable discussions and clarifications in the final version of the manuscript.

---

### Official Review · Reviewer_Qe43 · 2023-07-11

**Soundness:** 3 good
**Presentation:** 2 fair
**Contribution:** 3 good
**Rating:** 5
**Confidence:** 4

**Summary:**

This paper explores the concept of prompt tuning in the context of limited labeled data. The authors propose a unified objective function that encompasses three different learning paradigms, and investigate three distinct training strategies for leveraging pseudolabels. The experimental results on six datasets demonstrate the effectiveness of the proposed GRIP strategy. The findings of this study highlight the value of utilizing pseudolabels generated by CLIP itself to enhance its performance in diverse learning scenarios.

**Strengths:**

* this paper is well-written with a clear structure, making it easy to understand
* this paper extensively explores a board design space, including the prompt modality, the learning paradigm and the training strategy
* the authors conduct extensive expriments to verify the effectiveness of this method

**Weaknesses:**

* The writing of the article is not clear in some details, such as the typo error in line 290 and inconsistencies in abbreviations.
* please see questions listed below

**Questions:**

* I am little confused that does `line 217` means reinitializing the prompts at the begining of each iteration? Or does it mean only reinitializing the set of pseudolabels while the prompts are kept?
* Is the process of pseudolabeling done online or offline? If it is done offline, is it time-consuming when there is a large amount of unlabeled data?
* Are the abbreviations TZSL mentioned in `line 182` and TRZSL mentioned later referring to the same concept? This might be a bit confusing.
* The experimental results in Table 1 show significant improvements compared to CLIP or other baselines. However, there is a slight decrease in some cases. Perhaps the authors could try to explain the reasons behind this? For example, it could be related to the dataset or other factors.
* I think there might be a typo error in `line 290`. It should be Table 2 instead of Table 5.
* Have you validate the effectiveness on datasets with more categories (e.g, ImageNet-1K), because the quality of pseudo labels might decrease when the category space get larger.

**Limitations:**

I think the authors have adequately discussed the limitations of their research.

---

> ### Author Rebuttal · Authors · 2023-08-06
>
> Thank you for your thoughtful feedback. Below we address all your questions.
>
>
> > Does line 217 means reinitializing the prompts at the beginning of each iteration? Or does it mean only reinitializing the set of pseudolabels while the prompts are kept?”
>
>
> That’s correct. In line 217, we say that after each iteration we recompute the pseudolabels and then reinitialize the prompts (both in IFPL and GRIP). The reason behind this choice is to avoid accumulating errors from previous iterations.
>
> > Is the process of pseudolabeling done online or offline? If it is done offline, is it time-consuming when there is a large amount of unlabeled data?
>
> At each iteration, we generate pseudolabels for the unlabeled data from scratch. As the reviewer points out, the computation time becomes non-negligible with large amounts of unlabeled data.  We mitigated this issue via parallelization. We did not explore more sophisticated solutions, which we believe could become a topic of interest for future research, given the significant gain in performance obtained using pseudolabels dynamically.
>
> We thank the reviewer for raising this important question. We will add a brief discussion to the limitations section.
>
> > The experimental results in Table 1 show significant improvements compared to CLIP or other baselines. However, there is a slight decrease in some cases. Perhaps the authors could try to explain the reasons behind this?
>
> The few cases of slight decreases in performance in Table 1 happen for different reasons. For EuroSAT (SSL), the prompt modality in use matters. Indeed, adapting textual prompts with just a few labeled examples (CoOp) is enough to learn as much as we are able to learn with GRIP. This does not happens for visual prompts. The prompt modality has an impact also for Flowers102, indeed under the same settings, using textual prompts bring significant improvement. For FGVCAircraft, the initial set of pseudolabels has very low accuracy for SSL and UL (see Figure 6 in the Appendix) thus the learning process might become ineffective or even damage performance.
>
> > Have you validate the effectiveness on datasets with more categories (e.g, ImageNet-1K), because the quality of pseudo labels might decrease when the category space get larger.
>
> In our paper, we focused on datasets where CLIP performed poorly, primarily due to domain shift with the training data, as well as domain-specific tasks [31, Section 3.1.5, and Figure 5]. Consequently, we excluded larger datasets such as ImageNet-1K, which were either too similar to CLIP's training data distribution or represented very general domains. Exploring datasets with more categories could be an interesting avenue for future research.

---

> > ### Comment · Reviewer_Qe43 · 2023-08-18
> >
> > Thanks for the author's response, and I have already read all of the comments. My main concerns have already been solved, but I still believe that a scalable approach should have lower additional computational costs and be able to scale to datasets with a larger number of categories.

---

> > > ### Author Response · Authors · 2023-08-19
> > >
> > > Thank you for your response! We are glad that your main concerns have been solved. In light of that, we hope that you will consider increasing your support for our paper. If the scalability remains a concern, Reviewer kGE6 addressed this in their last comment:
> > >
> > > > About the computational budget: this argument as the weakness is not strong for me as there are a bunch of methods developed in many areas, like vision, MT and speech to have computationally efficient pseudo-labeling (when we train one model with regenerating pseudo-labels during training) -- so here I would not be surprised that they are applicable with small modifications and could resolve the computational budget issue.
> > >
> > > We agree that speeding up pseudo-labeling should not be a major challenge. CLIP even offers new tricks. For example, if only one prompt modality (text or image) is used, the embeddings of the other encoder can be cached. This will easily lead to an almost 2x speed up over a naive approach.
> > >
> > > Thank you so much for your valuable feedback.

---

### Author Rebuttal · Authors · 2023-08-06

We thank all the reviewers for their time and valuable feedback. Addressing your concerns during the discussion phase will significantly enhance the paper.

We clarify common questions here and address your reviews individually below.


### Novelty

Review qyTk expressed concern about the novelty of our training strategies using CLIP-based pseudolabels for prompt tuning. However, our paper's core novelty lies in the extensive exploration of an underexplored design space for prompt tuning with CLIP, which has been generally appreciated by all reviewers.

The design space includes prompt modalities, learning paradigms, and training strategies (Figure 1), that define 27 possible paths to explore, among which only 1 was previously investigated in literature [15]. We applied the same versatile training strategies to all the settings, based on the observations that (1) VLMs' zero-shot capabilities extend the usability of pseudolabels to any limited-label data scenario, and (2) paradigms such as semi-supervised, transductive zero-shot, and unsupervised learning can all be seen as optimizing the same loss function, by using zero-shot pseudolabels as a source of supervision.

While pseudolabeling with CLIP and pseudolabeling in general are not new, our perspective brings novel and meaningful contributions, particularly in the applicability of pseudolabels and their use to tailor CLIP for specialized domains with limited labeled resources.

### Naivety

While in our analysis we vary the prompt modalities, learning paradigms, and training strategies, we do not vary the rule for assigning pseudolabels and stick to the top-K strategy [15].  Reviewer sr7c criticized the usage of a too-naive rule. Although more sophisticated pseudolabeling and training strategies have been devised, we deliberately decided to stick to simple methods and have a better focus on the the three dimensions of the design space overviewed in Figure 1. It is impressive to see how this simple strategy leads to quantitative (impressive performance improvement) and qualitative advantages (Robin Hood effect), recognized by all reviewers.


### Clarification on training strategies

Reviewers Qe43 and kGE6 suggested to improve the clarity of the descriptions of  iterative strategies (IFPL and GRIP). In the manuscript, we will do that by adding explicit step-by-step explanations.

For IFPL, we begin by obtaining the top-K pseudolabels for each target class. These pseudolabels are then used to train a new task-specific prompt. After completing the training, we use the acquired prompt to compute the top-K pseudolabels per class again. Subsequently, we reinitialize the prompt and repeat this entire process for a total of I iterations.

As for the GRIP method, it shares similarities with IFPL, but with a key difference. In each iteration, we progressively increase the value of K. Specifically, during the ith iteration, we use K= i/I-th of the unlabeled data to perform the steps in the iterative process.


### Typos

We did our best to proofread the paper before the submission. However, we missed some minor typos which were highlighted by the reviewers, e.g., Table 5 in L290 should be Table 2. We resolved them. Thank you for pointing them out.

---

### Decision · Program_Chairs · 2023-09-21

**Decision:**

Accept (poster)

**Comment:**

This paper proposes to use CLIP for pseudo-labeling in various tasks such as SSL and on various datasets. Concerns were originally raised regarding unclear presentations, missing comparisons and discussion with alternative methods.

The authors provided a rebuttal and after the discussion period, 4 out 5 reviewers vote for acceptance. The AC believes the results are meaningful for future research using CLIP and recommends acceptance, and encourages the authors to revise paper accordingly.